# Fiber Bragg Grating Sensors: Design, Applications, and Comparison with Other Sensing Technologies

**DOI:** 10.3390/s25072289

**Published:** 2025-04-04

**Authors:** Alaa N. D. Alhussein, Mohammed R. T. M. Qaid, Timur Agliullin, Bulat Valeev, Oleg Morozov, Airat Sakhabutdinov

**Affiliations:** Department of Radiophotonics and Microwave Technologies, Kazan National Research Technical University Named after A. N. Tupolev-KAI, 10, K. Marx St., Kazan 420111, Russia; alaanajeed8181@gmail.com (A.N.D.A.); taagliullin@kai.ru (T.A.); bivaleev@kai.ru (B.V.); ogmorozov@kai.ru (O.M.); azhsakhabutdinov@kai.ru (A.S.)

**Keywords:** Fiber Bragg grating, structural health monitoring, sensing technology comparison, environmental and biochemical sensing, aerospace applications

## Abstract

Fiber Bragg grating (FBG) sensors have emerged as advanced tools for monitoring a wide range of physical parameters in various fields, including structural health, aerospace, biochemical, and environmental applications. This review provides a comprehensive overview of FBG sensor technology, focusing on their operating principles, key advantages such as high sensitivity and immunity to electromagnetic interference, and common challenges like temperature-strain cross-sensitivity and the high cost of interrogation systems. Additionally, this review compares FBG sensors with other sensing technologies and highlights recent innovations in design, packaging, and implementation techniques. Finally, future research directions are discussed to enhance the performance, scalability, and long-term reliability of FBG-based sensing systems.

## 1. Introduction

FBG sensors have become a widely adopted technology in optical sensing, particularly in structural health monitoring (SHM) [1], environmental monitoring [2], and aerospace applications [3]. FBG sensors operate by reflecting specific wavelengths of light in response to environmental changes.

Over the years, the development of FBG’s technology has progressed significantly. Early research focused primarily on optimizing the grating inscription process, improving sensitivity, and reducing cross-sensitivity between environmental factors such as strain and temperature. Later advancements incorporated packaging techniques to protect FBG sensors in harsh environments while also increasing their durability for long-term monitoring applications [4,5,6].

One of the key areas where FBG sensors have been extensively studied is in the field of SHM for civil infrastructure. Researchers such as Rodrigues et al. [5] and Braunfelds et al. [1] explored the use of FBG sensors in monitoring the behavior of bridges, dams, and buildings under various loading conditions. These studies demonstrated the ability of FBG sensors to accurately measure strain, displacement, and temperature changes in real time, which are critical for assessing the integrity of structures. Additionally, the use of FBG sensors in earthquake monitoring systems, as explored by López-Castro et al. [7], showcased the sensors’ ability to track dynamic responses of structures under seismic activity. Moyo et al. [8] demonstrated the use of FBG sensors for continuous monitoring of concrete bridges. Their research highlighted the durability of FBG sensors, even in harsh weather conditions, and their ability to provide accurate data over extended periods. This work was followed by Kuang et al. [9], who developed a long-term monitoring system based on FBG sensors to detect changes in stress and strain on concrete bridge decks.

In the context of environmental monitoring, FBG sensors are sensitive to environmental factors, particularly temperature and moisture, which can influence their performance. Therefore, packaging and protective coatings have become a significant area of research. Studies by Lin et al. [10] focused on developing advanced packaging techniques to enhance the robustness of FBG sensors for use in harsh environments, such as underwater or industrial applications. These studies provided innovative solutions for embedding FBG sensors in composite materials or encasing them in protective coatings that minimize degradation due to environmental exposure.

A major area of research has been comparing FBG sensors with other existing sensing technologies. Wang et al. [11] conducted a comprehensive study on the design and verification of FBG strain gauges, electromagnetic interference (EMI) immunity, and the potential for distributed sensing over long distances. Further comparisons, such as those conducted by Pashaie et al. [12], illustrated the advantages of FBG sensors over other optical fiber sensors, such as interferometric sensors, in terms of scalability and ease of integration into existing monitoring systems.

The versatility of FBG sensors has also been explored for environmental and biochemical sensing applications. Researchers such as Bui et al. [13] proposed a novel method based on dual FBGs integrated into a fiber ring laser structure for biochemical sensing applications. Moreover, Yulianti et al. [14] conducted an in-depth investigation into the application of FBG sensors for pH measurement.

FBG sensors have also found applications in the aerospace industry, where precision and reliability are crucial. Studies by Kahandawa et al. [15] and Hegde et al. [16] investigated the use of FBG sensors for monitoring strain, temperature, and pressure in aircraft structures and satellite systems. The lightweight, compact nature of FBG sensors, combined with their resistance to electromagnetic interference, makes them ideal for aerospace applications where space and weight are at a premium.

Figure 1 illustrates the development in the number of research publications related to FBG sensors from the early 1970s to the present.

This review paper aims to provide a comprehensive examination of the current state of research, and applications of FBG sensors across various domains, and suggest directions for future research that could address current challenges and expand the applicability of FBG sensors in emerging fields. By evaluating the advancements in sensor design, implementation methods, and packaging techniques, we will assess the effectiveness of FBG sensors in SHM, environmental sensing, biochemical applications, and aerospace technology. In addition, this paper compares FBG technology with other sensing technologies to highlight its strengths and limitations in different contexts. Finally, we will discuss the long-term monitoring capabilities of FBG sensors, particularly in the context of concrete bridge monitoring, to provide insights into their role in infrastructure sustainability.

## 2. FBG Technology Fundamentals

An optical fiber consists of three main parts: the core, the cladding, and the protective jacket. The central part of the fiber, where the Bragg grating is inscribed, is typically made of doped silica. This core guides light and allows the Bragg grating to reflect specific wavelengths. Surrounding the core, the cladding has a lower refractive index (RI), helping to confine the light within the core and ensuring efficient light transmission with minimal loss. The outermost layer, or jacket, protects the cladding and core from physical damage, environmental factors, and mechanical stress, enhancing the sensor’s durability.

This section provides a detailed explanation of the FBG working mechanism, where shifts in the Bragg wavelength occur in response to environmental changes, such as strain, temperature, etc. It also describes the grating inscription process, calibration procedures, and the key advantages of FBG technology, such as high sensitivity and multiplexing capabilities, alongside limitations, including temperature cross-sensitivity and the cost of interrogation systems.

### 2.1. Principles of Operation

FBG sensors operate based on the Bragg diffraction principle, where specific wavelengths of light are reflected back when they interact with a grating—a periodic variation in the refractive index along the fiber core, which is illustrated in Figure 2.

The Bragg wavelength, $\lambda_{B}$_,_ is a specific wavelength of light that is reflected by the FBG and is defined by the following formula:(1)$$\lambda_{B}=2\cdot n_{e\mathrm{ff}}\cdot \Lambda$$
where $n_{e\mathrm{ff}}$ is the effective refractive index of the fiber core, and Λ is the grating period or spacing between the refractive index variations.

Changes in strain or temperature alter $n_{e\mathrm{ff}}$ and/or Λ, leading to a shift in the Bragg wavelength ($\Delta \lambda_{B}$). This shift can be measured to determine the magnitude of the applied stimulus:(2)$$\Delta \lambda_{B}=2\cdot \left(\frac{\partial n_{e\mathrm{ff}}}{\partial T}\Delta T+n_{\mathrm{eff}}\frac{\partial \Lambda }{\partial \varepsilon }\Delta \varepsilon \right)$$
where ΔT is the temperature change, and Δε is the strain change.

FBG can also be employed to measure acidity by applying a specialized coating to FBG. When the sensor’s temperature remains constant, the stress induced in the pH-sensitive gel layer causes a shift in the Bragg wavelength. Consequently, the Bragg wavelength may be expressed as follows [18]:(3)$$\frac{\Delta \lambda_{B}}{\lambda_{B}}=K_{\varepsilon }\times \varepsilon$$
where ε is the axial stress of the FBG, which is caused by the expansion or contraction of the pH-sensitive gel layer coated on the FBG, and $K_{\varepsilon }$ is constant.

Cladding-etched fibers can be used to measure the refractive index of the surrounding medium since the FBG itself is not sensitive to the refractive index of the surrounding medium. These sensors are based on increased evanescent field interaction with the measurand, and they have found implementations in chemical and biological fields to measure various liquids and gases. Cladding etching results in a progressive change of the effective refractive index of the propagating mode due to increasing mode confinement. The Bragg wavelength shifts as a function of the refractive index of the ambient material. FBG based on single-mode fiber supports the fundamental mode. During cladding etching, higher-order Bragg resonances appear when the fiber diameter reduces. These higher-order Bragg resonances are used to determine the diameter of a standard optical fiber with a precision of ~200 nm. The evanescent field values of the fundamental mode and higher-order modes are affected by the surrounding refractive index to different extents. Using the relative wavelength shifts between the modes, the refractive index of external media can be determined. FBGs were etched using buffered hydrofluoric acid and then immersed in liquids such as methanol, ethanol, isopropanol, and ethylene glycol [19,20].

This mechanism makes FBG sensors highly suitable for monitoring a wide range of physical parameters. The wavelength shift is inherently linear and can be precisely measured, allowing for high-resolution sensing.

FBG sensors are typically inscribed using ultraviolet (UV) light to create periodic changes in the refractive index of the fiber core. However, femtosecond ($f_{s}$) laser technology has revolutionized FBG inscription by enabling the fabrication of gratings in various fiber types, including those insensitive to light, without requiring hydrogen loading. $f_{s}$ laser inscription methods, such as phase mask, point-by-point, and line-by-line techniques, offer flexibility and precision. Unlike UV-based methods, $f_{s}$ lasers eliminate the need for hydrogen-loading and can directly inscribe FBGs in standard fibers while also allowing for the creation of complex structures like accurately apodized FBGs [21,22,23,24]. Key methods of FBG inscription include:

Phase Mask Technique: A UV laser is shined through a phase mask to form the grating pattern. This technique is common due to its simplicity and precision [25,26].

Point-by-Point Writing: The UV laser directly writes the grating into the fiber, allowing for customization of the grating pattern but requiring more complex equipment [27,28].

Interferometric Technique: Interference patterns are used to inscribe the grating, typically providing higher efficiency and uniformity [29].

FBGs can also be classified by the conditions of mode coupling formation, such as homogeneous, chirped, and with phase shift.

Homogeneous FBGs: These have a uniform grating period along the entire length. They are simple to fabricate and commonly used in standard applications such as temperature and strain sensing. However, they may have limited flexibility for complex or broadband sensing [30].

Apodized FBGs: These gratings have a tapering intensity profile, where the grating’s intensity decreases toward the edges. This design helps minimize the sidelobe intensity in the reflection spectrum, making them particularly advantageous for applications requiring high spectral purity. Apodized FBGs are commonly utilized in interferometric methods due to their superior spectral characteristics compared to uniform (non-apodized) gratings, which exhibit stronger sidelobes in the reflection spectrum [31,32].

Chirped FBGs: The grating period varies along the length of the fiber, allowing for a wider range of applications, including dispersion compensation and broadband sensing. These FBGs offer higher sensitivity but may be more challenging and expensive to fabricate [33].

FBGs with Phase Shift: Phase-shifted FBGs introduce a phase discontinuity in their structure, enabling precise filtering and sensing applications like vibration monitoring [34]. This phase shift is achieved by altering the grating’s pitch, often through selective cutting. While enhancing spectral resolution and sensitivity, these FBGs involve complex fabrication processes and may have lower reflectivity compared to standard FBGs.

FBGs are traditionally inscribed using UV lasers; however, recent advancements have demonstrated that femtosecond lasers are more effective for FBG inscription. This method, as investigated in [35,36], offers certain advantages, such as the ability to inscribe in non-photosensitive fibers and achieve higher inscription efficiency in specialized applications. The inscription of gratings with one or multiple phase shifts is best performed using the point-by-point inscription method [37]. A comparison of these methods is beyond the scope of this study but is briefly noted here to provide a broader context for the reader.

In addition, FBGs are categorized into Types I and II based on their photosensitivity and response to UV exposure [38]:

Type I FBGs: These are the most common and are formed by low-intensity UV exposure. They offer high sensitivity and stability, making them ideal for a wide range of sensing applications, such as temperature and strain measurements [36].

Type II FBGs: These are created by higher UV intensities and exhibit higher thermal stability and resistance to radiation. They are particularly useful in high-temperature or extreme environmental conditions but are more difficult and expensive to manufacture.

Each type of FBG has its advantages and disadvantages in sensor applications, depending on the required sensitivity, stability, and operating conditions.

The choice of inscription method affects the sensor’s performance characteristics, such as reflectivity, grating length, and stability. Calibration is essential to ensure the accuracy and repeatability of FBG measurements. Calibration procedures typically involve:

Strain Calibration: The sensor is exposed to known levels of strain to create a reference curve for wavelength shift versus strain [39].

Temperature Calibration: The sensor is subjected to controlled temperature changes to determine the relationship between temperature variations and the Bragg wavelength shift [39].

Cross-Sensitivity Correction: Since temperature and pressure influence the Bragg wavelength, calibration helps to establish compensation methods to isolate the desired measurement variable. Since FBG has combined sensitivity to both temperature and pressure simultaneously, it is necessary to use calibration methods for combined pressure and temperature sensors [40].

### 2.2. Advantages and Limitations of FBG Sensors

FBG technology offers numerous advantages that make it a valuable sensing tool across multiple fields, though it also encounters some specific challenges. The technology also supports strong multiplexing capabilities, allowing multiple FBGs to be inscribed along a single fiber [41]. As already mentioned, FBGs are resistant to EMI, which makes them especially advantageous in environments with high electromagnetic noise, like power plants and substations [42]. The flexible design of FBGs allows for embedding into materials without altering their properties, providing significant benefits for civil and aerospace engineering [15,43]. This quality ensures that their performance is less affected by fluctuations in light sources or fiber loss, further enhancing measurement reliability over time [44].

Despite offering stable and accurate measurements due to their reliance on wavelength shifts rather than intensity changes, FBG sensors are not without challenges. One significant drawback is the high cost of interrogation systems, which are necessary to detect and analyze the wavelength shifts. This can limit the scalability of FBG systems, especially in projects with tight budgets [44]. Another issue is temperature cross-sensitivity; temperature variations impact the Bragg wavelength, potentially interfering with strain measurements. Compensation methods, such as using additional reference sensors, are often required to accurately isolate strain and temperature data [45,46]. Durability in harsh environments presents additional challenges. In [47], the need for effective packaging of FBG sensors for offshore structural health monitoring, particularly in floating, production, storage, and offloading units, is highlighted. After thorough research, optimal commercial packaging was selected and tested against harsh marine conditions strong sunlight, heavy rain, and saltwater to ensure long-term sensor reliability. Additionally, while FBG sensors offer multiplexing capabilities, the number of sensors that can be deployed along a single fiber is limited by the available wavelength range. This restriction necessitates careful planning in terms of sensor spacing and distribution to maximize their effectiveness across a structure [48].

## 3. Design, Implementation, and Packaging Techniques

In the deployment of FBG sensors, the design, implementation, and packaging are crucial for achieving accurate and reliable measurements. These steps influence the sensors’ performance, durability, and suitability for different applications. Here, we detail each aspect, emphasizing the critical parameters, advanced coating and packaging techniques, and various implementation methods to enhance FBG sensor performance in structural and environmental monitoring.

### 3.1. Sensor Design

FBG sensor design impacts performance factors such as sensitivity, precision, and durability. The key design parameters that influence FBG performance are as follows [49,50,51]:

Grating Length: The length of the grating determines the sensor’s wavelength reflectivity and spectral width. Longer gratings offer improved sensitivity, while shorter gratings may be more suitable for high-resolution applications due to narrower spectral peaks.

Core Diameter: The core diameter affects the sensitivity and bending tolerance of the FBG sensor. Smaller core diameters can increase sensitivity but may reduce mechanical robustness.

Refractive Index Modulation: The index modulation determines the strength of reflection and wavelength stability. Higher modulation allows stronger reflection, which can improve signal clarity.

Developments in FBG coatings that can help improve sensitivity and stability can be summarized as follows [52,53]:

Polymer Coatings: By using polymers with specific thermal and mechanical properties, FBG sensors can achieve higher sensitivity, particularly for strain and temperature variations.

Metal Coatings: Metal coatings, such as gold or aluminum, are applied to FBG sensors to protect the sensor’s metal components from corrosion and environmental degradation, rather than the quartz glass optical fiber itself. While quartz glass is highly resistant to corrosion and remains stable under most harsh conditions, the metal coatings help ensure the overall durability of the sensor, especially in challenging environments. These coatings shield sensitive parts of the sensor from external factors like moisture, chemicals, or physical wear that could impact its performance over time, thereby extending the sensor’s lifespan and reliability [54].

Hydrophobic Coatings: These coatings improve stability by reducing moisture interference, which is essential for FBGs exposed to high humidity or submerged in liquids. Table 1 provides an overview of key design parameters that influence the performance of FBG sensors, highlighting their impacts and relevant example applications.

### 3.2. Implementation Techniques

The deployment method of FBG sensors varies depending on the structure and environment, including embedding sensors in concrete or attaching them to structural surfaces. FBG deployment techniques can be discussed in the following [59].

Embedding in Concrete: FBGs can be embedded directly into concrete during the pouring phase, allowing them to monitor internal parameters such as temperature, strain, and pressure within the structure. This method is beneficial for long-term monitoring and minimizes sensor visibility.

Surface Attachment: In this technique, FBGs are attached to the surface of structural elements using adhesives or mounting brackets. This approach is preferable for retrofitting existing structures and provides accurate surface strain measurements.

### 3.3. Packaging Techniques

Protective packaging is essential to maintain FBG sensor effectiveness, particularly in harsh environments. Effective packaging techniques help ensure the long-term durability and consistent performance of the sensors, safeguarding them from environmental and mechanical stresses.

Hermetic sealing is one of the primary methods used to protect FBG sensors. It involves enclosing the sensor in a moisture-proof barrier, typically made of glass or metal, to prevent moisture ingress. This technique is particularly important for sensors used in marine or underground applications, where high humidity levels could damage the sensor [60]. Another widely used method is applying metal and polymer coatings. These coatings protect the FBG sensors from physical wear, corrosion, and chemical exposure. For instance, a stainless-steel sleeve with a polymer coating provides robust protection, making it suitable for industrial and chemical environments [61]. Additionally, shock-absorbing materials like rubber or silicone are added to protect the sensors from mechanical impacts. This packaging is particularly crucial in applications such as transportation or infrastructure, where sensors are exposed to frequent vibrations [62].

Despite their protective benefits, packaging techniques also introduce some challenges. One issue is thermal mismatch, where differences in thermal expansion between the sensor and packaging materials can create strain that affects the sensor’s readings [63]. Another challenge is the cost and complexity associated with advanced packaging techniques, such as hermetic sealing. These methods increase production costs and may be difficult to implement for large-scale applications [64]. Moreover, packaging can add extra weight and size to the sensor, which may limit its use in confined spaces or in applications that require lightweight sensors.

Table 2 presents the refractive index sensitivity, ranges, and key additional details from each reference.

In recent years, additive manufacturing (AM) has emerged as a promising technique for embedding FBG sensors. AM-based approaches allow for the precise integration of FBGs into complex structures, enabling tailored sensor applications. For instance, Kuang et al. [69] demonstrated the potential of embedding FBGs in composite materials, laying the groundwork for advanced packaging methods. More recently, Yan et al. [70] introduced a 3D printing-based online packaging technology for FBGs, showcasing the capability of additive manufacturing to enhance sensor integration and functionality. Zhao et al. [71] utilized ultrasonic additive manufacturing for dynamic strain measurements, providing insights into high-frequency applications. Paloschi et al. [72] further evaluated 3D-printed sensing samples with embedded FBGs, focusing on metrological aspects across various materials and coatings.

## 4. Applications Across Domains

### 4.1. Displacement Measurement

The sensors are usually embedded within structures or attached to measure relative displacement across key points. Table 3 provides a comparison of FBG sensors for displacement monitoring across various sectors, highlighting their measurement range, precision, environment, and benefits:

### 4.2. Liquid Level Monitoring

FBG sensors are adapted for liquid-level detection, especially in storage tanks and reservoirs. They measure changes in pressure or strain as the liquid level fluctuates. Table 4 summarizes the use of FBG sensors for liquid-level monitoring in various applications.

Figure 3 illustrates the practical applications of FBG sensors across various engineering fields, providing a comprehensive view of their versatility and utility. In Figure 3A, the use of FBG sensors in civil engineering is showcased, where they are installed on slope structures to monitor displacement, ensuring early detection of potential failures [84]. Figure 3B highlights their application in aerospace engineering, where FBG sensors are embedded in aircraft wings for structural health monitoring, enhancing safety and performance [85]. Figure 3C presents their role in biomedical engineering, specifically in wearable devices for tracking human motion, supporting advancements in healthcare and rehabilitation technologies [86]. Lastly, Figure 3D illustrates an energy harvesting setup using a FBG temperature sensor. A broadband amplified spontaneous emission source from the erbium-doped fiber amplifier travels through a circulator and interacts with the FBG, whose reflected light is analyzed by the optical spectrum analyzer to measure temperature shifts in the Bragg wavelength. The unreflected portion of the optical signal passes through the FBG into an energy harvesting unit. The harvested optical power is converted into electrical energy by a photodiode operating in photovoltaic mode, along with a series resistor, shunt resistor, and junction capacitor. This circuit allows simultaneous temperature sensing and optical power harvesting [87].

### 4.3. Temperature and Pressure Sensing

The Bragg wavelength shift allows for precise measurements of temperatures and pressure parameters. Table 5 provides an overview of FBG sensors used for temperature and pressure sensing in different applications.

### 4.4. Strain Monitoring

FBG sensors are recognized for their remarkable accuracy in quantifying strain across a wide range of structural configurations. Table 6 outlines the use of FBG sensors for strain monitoring in various applications.

FBG sensors and Brillouin Optical Time Domain Reflectometry (BOTDR) sensors are both optical fiber-based sensing technologies used for strain measurement, but they operate on different principles and have distinct advantages depending on the application [94,95].

FBG sensors offer high spatial resolution and precision but are best suited for localized strain measurements. On the other hand, BOTDR sensors are more appropriate for distributed sensing across large areas or structures, making them ideal for continuous monitoring of strain over long distances.

### 4.5. Environmental and Biochemical Applications

FBG sensors are increasingly employed to monitor essential parameters, like pH, glucose levels, and humidity, and to detect various biochemical agents [96,97].

Chromium is a harmful metal found in industrial effluents. Mondal et al. [98] demonstrated the use of FBG sensors coated with polyvinylpyrrolidone-functionalized silver nanoparticles for the detection of chromium (VI) in drinking water. The sensor exhibits high sensitivity, with a detection limit of 0.1 pM (0.0000052 ppb), well below the World Health Organization permissible limit. FBG sensors can also measure the concentration of red blood cells, which helps in detecting blood-related diseases. Yao et al. [99] proposed a graphene-based D-shaped polymer FBG for highly sensitive erythrocyte detection. The sensor utilizes p-doped graphene, which enhances the evanescent field for improved sensitivity. It detects erythrocyte solutions with concentrations ranging from 0 to 10⁴ ppm, achieving sub-ppm resolution. FBG sensors also can be functionalized to detect specific biochemical markers, such as pH and glucose [100]. Table 7 provides a comprehensive overview of the various applications of FBG sensors in environmental and biochemical sensing.

## 5. Structural Health Monitoring Applications

FBG sensors have proven effective in numerous SHM projects worldwide, providing critical data that enable proactive maintenance and safety measures. FBG sensors have demonstrated exceptional efficiency in detecting cracks and monitoring structural health. The FBG sensors were fixed on the lower surface of a beam to capture strain and detect the initiation of microcracks in real time. The study found that FBG sensors could accurately monitor surface crack formation and even estimate crack widths with reasonable precision [105]. FBG sensors were deployed on various bridges to monitor strain, displacement, and temperature fluctuations [106,107]. FBG sensors were utilized in both high-rise buildings [93], smart civil structures [108], and dam infrastructures. For high-rise buildings, these sensors monitored seismic effects, helping to assess the buildings’ resilience against earthquakes. FBG sensors are used in dams to monitor strain, internal displacement, and structural responses during dynamic events such as earthquakes. They are applied in small-scale gravity dam models to assess seismic behavior, detect cracks, and evaluate overall structural safety [109,110,111]. Similarly, FBG sensors have been applied in tunnel monitoring projects [112]. These sensors provided continuous assessments of displacement, convergence, and deformation within tunnels, contributing to improved structural integrity over time and aiding in accident prevention. This application illustrates FBG technology’s role in enhancing tunnel safety and performance through ongoing structural assessment [106].

### 5.1. High-Rise Building

High-rise buildings (over 40 floors or 100 m) demand more sophisticated structural health monitoring due to their heightened safety requirements. The advancement of intelligent materials has introduced innovative strategies for SHM of high-rise buildings. FBGs, due to their small size, are regarded as ideal sensing elements for creating intelligent composite materials. In [113], FBG sensors were embedded into intelligent composite materials within high-rise buildings, enabling the monitoring of key parameters such as strain and temperature. By integrating FBG directly into these materials, the article shows how it becomes part of the building’s structure. In [114], a total of 120 FBG sensors were installed on the main tower, arranged in five groups of 24 sensors each. Every group is further divided into four arrays, each containing six sensors: four for measuring dynamic strain and two for temperature. An additional 80 sensors were mounted on the antenna mast, where 64 measure strain and 16 measure temperature. These sensors are welded onto steel-tube columns at two levels—one above and one below the welding line—to capture strain data on opposite sides of the columns. They are packaged in stainless-steel for protection and longevity, enabling direct welding to the structure. The sampling rate for dynamic strain measurement was set to 50 Hz.

### 5.2. Long-Term Monitoring of Concrete Bridges

Long-term monitoring of concrete bridges is essential to ensure their structural integrity, safety, and lifespan. The advantages of FBG in long-term monitoring are well-suited for the long-term monitoring of bridges because they are constructed from optical fibers, which are resistant to harsh environmental conditions. This durability allows FBG sensors to perform reliably over extended periods. Their high stability and precision also make them dependable for continuous monitoring without significant drift or sensitivity loss. Additionally, FBG sensors support multiplexing, allowing multiple sensing points along a single fiber ideal for monitoring large structures like bridges. This capability reduces installation costs and simplifies data collection processes. Furthermore, unlike traditional sensors that may require regular recalibration or replacement, FBG sensors generally maintain stable performance with minimal maintenance.

Numerous case studies illustrate the effectiveness of FBG sensors in concrete bridge monitoring. For example, FBG sensors were installed on a bridge to track strain, temperature, and load distribution over time [115,116,117,118]. In [119], FBG sensors, integrated with a hydrostatic leveling system, measure vertical displacements on bridges and require no external reference. Laboratory and field tests confirm their accuracy for both short- and long-term structural monitoring.

FBG sensors detect early cracks and damage, preventing catastrophic structural failures [120]. For instance, in reinforced concrete beams subjected to cyclic loading, FBG sensors embedded along the critical tension zones can measure localized strain variations. These sensors are highly sensitive to micro-strain changes, enabling the identification of microcracks before they propagate into larger, more hazardous fractures. In one application, FBG sensors detected strain concentrations in a bridge deck under traffic loads, signaling the need for immediate maintenance. This early warning system provided sufficient time to implement corrective actions, thereby preventing structural failure and ensuring public safety [121].

Another example involves the use of FBG sensors in steel truss bridges where strain and displacement monitoring were crucial during extreme weather conditions. Measurements from the sensors revealed abnormal stress concentrations in specific joints caused by thermal expansion and contraction [107]. By analyzing the real-time data, engineers identified potential weak points and reinforced them before damage could occur. These examples underscore the importance of FBG sensors in not only detecting structural anomalies but also providing actionable insights that allow for timely interventions, enhancing the resilience and safety of critical infrastructure.

## 6. Comparison with Other Sensing Technologies

FBG sensors are often compared to other sensing technologies, such as electronic sensors, microelectromechanical systems (MEMSs), and other optical fiber sensors. Each technology has strengths and limitations. Here, we delve into the comparative advantages and applications of FBGs relative to these other technologies, using key parameters such as sensitivity, durability, environmental resilience, and cost.

### 6.1. FBG vs. Electronic Sensors

Electronic sensors, including resistive and capacitive sensors, are commonly used for monitoring physical parameters such as strain, temperature, and displacement. However, FBG sensors offer distinct benefits over traditional electronic sensors, particularly in environments where EMI and durability are concerned.

FBGs’ abilities to deliver reliable data in challenging environments justifies their higher cost and positions them as the preferred choice for precision monitoring and long-term structural integrity assessments. Electronic sensors, although less robust and sensitive, remain valuable in general-purpose monitoring where budget constraints and lower environmental demands are key considerations. Their lower cost and adequate performance make them appropriate for widespread, routine applications where high sensitivity and extreme durability are not essential. Ultimately, FBG sensors provide a highly capable and reliable option for complex monitoring needs in high-stress environments, while electronic sensors serve as an affordable and practical solution for broader, less demanding applications. The choice between these sensors thus depends on the specific demands of the application, balancing factors like cost, durability, and measurement precision.

### 6.2. FBG vs. MEMS Sensors

Microelectromechanical system sensors are compact and commonly used in applications requiring small-scale sensors, such as wearable devices or consumer electronics. FBGs, however, offer higher precision and are more resilient in extreme environments, making them suitable for industrial and structural applications. Table 8 provides a detailed comparison between FBG and MEMS sensors across several key parameters, including sensitivity, measurement range, response speed, temperature limits, uncertainty, and overall size, highlighting how each sensor type aligns with specific application needs. This table is intended to assist in understanding the distinct advantages and limitations of FBG and MEMS sensors, enabling informed decision-making in various high-precision and industrial contexts.

### 6.3. FBG vs. Other Optical Fiber Sensors

Table 9 presents a concise comparison of key performance parameters, such as temperature sensitivity, pressure sensitivity, and strain sensitivity, for various types of optical fiber sensors. These include FBG sensors, interferometric sensors, long-period grating sensors, and distributed sensors.

## 7. Challenges and Future Directions

Despite the advantages that FBG sensors possess, they still face several critical challenges before they can be widely deployed. Among these are cross-sensitivity issues that can interfere with measurement accuracy, the relatively high cost of interrogation systems, and questions regarding long-term durability. There is also a need for further advancement in miniaturization and multiplexing to optimize their performance.

### 7.1. Cross-Sensitivity Issues: Temperature-Strain Interaction in FBG Sensors

One of the primary technical challenges in FBG sensing is cross-sensitivity, where the measurement of one physical parameter (such as strain) can be influenced by another parameter (like temperature) [45]. Both strain and temperature affect the Bragg wavelength, making it difficult to separate their individual impacts in environments where both parameters fluctuate. This cross-sensitivity can compromise the accuracy of measurements, especially in critical applications where precise data are required [159].

To mitigate these issues, a dual FBG system, with one sensor placed under strain and another as a reference (unstrained), makes it possible to separately measure temperature and strain. The reference sensor compensates for temperature changes, ensuring a more accurate reading of strain. Integrating FBGs with dedicated temperature sensors allows for independent measurement of temperature and strain, which can greatly improve accuracy. Advanced coatings can be applied to FBGs to selectively enhance their sensitivity to either temperature or strain. These coatings help reduce the cross-sensitivity effect, but they come with durability concerns. In [160], researchers addressed the issue of temperature-strain cross-sensitivity in FBG-based sensors by developing an advanced sensor known as a super-structured FBG sensor, where periodic metal thin films were deposited on the fiber surface. Although traditional methods using dual FBGs can reduce cross-sensitivity when measuring multiple parameters, these approaches are often complex. Recently, Artificial Neural Networks (ANNs) have been introduced as an alternative, offering a simpler and more cost-effective solution to minimize cross-sensitivity in FBG sensors. In [161], the researchers developed an accurate method for the simultaneous measurement of strain and temperature using FBG sensors. To address the challenge of cross-sensitivity, they integrated an ANN model with an etched FBG sensor combined with a Single-Multi-Single fiber structure. Experimental data were collected to train and optimize the ANN for enhanced performance. The proposed system demonstrated significantly improved sensitivity and reduced measurement error compared to standard matrix inversion techniques. Specifically, temperature and strain sensitivities increased by 1.7 and 2 times, respectively, while root mean squared errors were reduced by factors of 680 for temperature and 164 for strain.

### 7.2. Cost of Interrogation Systems: Strategies for Cost Reduction

FBG sensors require high-performance interrogation systems to extract meaningful data. These systems analyze the Bragg wavelength shifts, which can be expensive, making FBG applications prohibitively costly, especially in large-scale systems. For FBG technology to achieve broader adoption, especially in long-term monitoring and large networks, cost reduction is essential.

Several cost-reduction strategies have been explored, including placing multiple FBG sensors along a single optical fiber to reduce the need for multiple interrogation systems. This technique not only lowers costs but also simplifies the overall sensor setup [162]. By designing interrogation systems that can handle multiple FBG arrays, the cost per sensor can be significantly reduced. This approach is particularly useful in large-scale sensor networks. Additionally, the development of smaller, affordable spectrometers could replace expensive interrogation systems in less demanding applications. This would reduce the overall cost of the system but could compromise performance in high-precision environments.

### 7.3. FBG Interrogation Methods

FBG interrogation methods are crucial for extracting data from the sensors by detecting shifts in the Bragg wavelength. These methods can vary in complexity, sensitivity, and cost. Table 10 presents an overview of the most common FBG interrogation techniques, along with their cost implications.

### 7.4. Hybrid FBG-MEMS Devices

The FBG-MEMS technology combines FBG sensors and microelectromechanical systems to monitor stress in soils, foundations, and structures. FBG sensors detect shifts in reflected wavelengths caused by axial strain, providing high sensitivity in measuring stress and temperature. MEMS sensors use the piezoresistive effect, converting applied stress into measurable electrical voltage. Experiments showed FBG-MEMS accurately measures three-dimensional soil stresses, including shear and principal stresses, and identifies principal stress directions. This technology has been successfully used in static pressed pile tests, clarifying pile penetration mechanisms and pile–soil interface stresses. However, these technologies can only monitor the normal stress in one direction and cannot monitor the shear stress and the principal stress [167,168,169].

### 7.5. Enhancement of Sensing Elements in Surgical Robots

FBG sensors are increasingly being utilized in surgical robots to enhance force sensing, improve precision, and ensure safety during minimally invasive surgeries (MISs). One notable application of FBG technology is in catheter surgical robots, where FBG sensors are used to monitor proximal forces, such as clamping and axial forces. These sensors, integrated into the robot’s gripper, provide real-time force feedback, ensuring safe and accurate catheter movement during vascular interventions. The sensors offer high resolution (<2.5 mN) and sensitivity (>480 pm/N) [170]. In endoscopic surgical robots, FBG sensors are used to measure the compression force exerted on the sheath of tendon–sheath mechanisms, providing a non-invasive way to infer the tension on the tendon. This application is particularly useful in tendon-driven systems, such as robotic hands and surgical catheters, where small size and flexibility are critical [171]. In orthopedic surgery robots, a six-dimensional FBG force/moment sensor has been developed to detect the interaction forces between a drill and bone during robot-assisted bone drilling [172]. Furthermore, sensorized surgical instruments based on FBG technology have been introduced to improve force measurement capabilities in MIS [173].

Despite the increasing interest in FBGs in scientific research and their potential medical applications, there remains a significant gap between research and clinical practice. In fact, FBG-based devices are still almost non-existent in clinical settings. Therefore, it is essential to further develop the working principles and designs of FBG sensors to make them more practical and suitable for real-life applications.

## 8. Conclusions

FBG sensors offer important benefits such as high sensitivity, immunity to electromagnetic interference, and suitability for harsh environments. These attributes make them valuable for applications ranging from structural health monitoring to biochemical and environmental sensing. However, several challenges still need to be addressed to enable more widespread adoption. Chief among these are the cost of high-performance interrogators, the potential interference between temperature and strain measurements, and the complexities associated with robust packaging and protective coatings. Ongoing research into multiplexing techniques, miniaturization, and integration with emerging technologies—such as MEMSs and the Internet of Things—will be critical for improving cost-effectiveness, scalability, and long-term reliability. As these challenges are addressed, FBG sensors will become increasingly adaptable, enabling a broader range of practical applications.

## Figures and Tables

**Figure 1 sensors-25-02289-f001:**
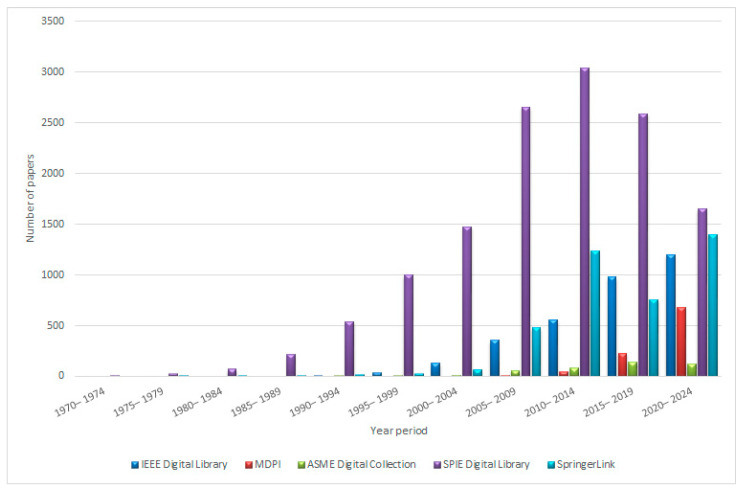
Number of research papers from 1970 to 2024 using the search terms “FBG sensor”.

**Figure 2 sensors-25-02289-f002:**
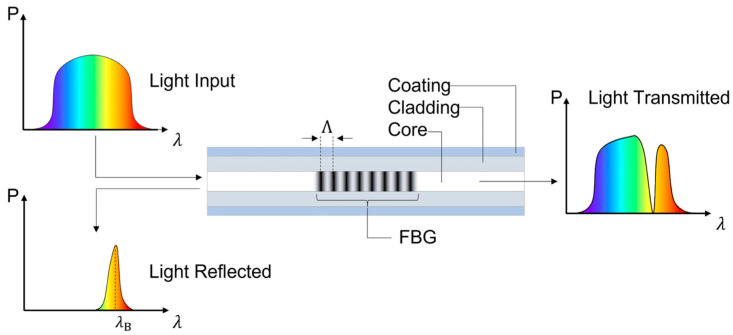
The working principle of an FBG as a sensor [17].

**Figure 3 sensors-25-02289-f003:**
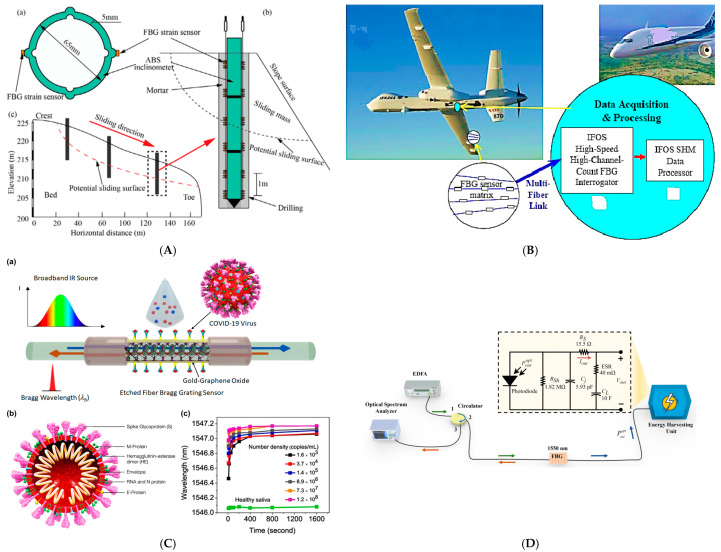
Real-world applications of FBG fiber sensors in various engineering fields. (**A**) FBG application in slope structures: (a) cross-section; (b) longitudinal section; (c) diagram of installation [84]. (**B**) FBG application in aerospace engineering [85]. (**C**) FBG application in biomedical engineering: (a) Schematic diagram of the sensing mechanism of FBG; (b) Schematic diagram of COVID-19; (c) Wavelength of detected light versus exposure time for various virus concentrations [86]. (**D**) FBG application in energy systems [87].

**Table 1 sensors-25-02289-t001:** Design parameters and their effects on FBG performance.

Parameter	Impact on Performance	Example Applications	Ref.
Grating Length	Shorter grating lengths (e.g., 2 mm) improve sensor stability and accuracy under strain gradients compared to longer lengths (e.g., 10 mm)	High-precision strain monitoring	[55]
Diameter	The small diameter enhances the sensitivity	Sensitive pressure measurements in medical application	[56]
Polymer Coatings	Increases humidity response	Humidity, temperature applications	[53,57]
Metal Coatings	Pressure sensitivity increased 19 times and temperature sensitivity doubled; measurable up to 40 MPa and 240 °C	Suitable for harsh environments such as oil wells monitoring	[58]

**Table 2 sensors-25-02289-t002:** Overview of refractive index sensitivity and packaging techniques for optical sensors.

Sensor Type	Refractive Index Sensitivity	Refractive Index Range	Packaging Technique	Ideal Applications	Potential Challenges	Ref.
Etched FBG in Panda fiber	−1.492 nm/RI-unit (RIU)	1.333 to 1.443 RIU	Chemical etching of the cladding	Temperature-independent refractive index sensing	Increased radiation loss	[65]
Optoelectronic oscillator with etched phase-shifted FBG	530 MHz/RIU	1.341 to 1.350 RIU	Cladding-etched PS-FBG	Biochemical and chemical RI sensing	Cross-sensitivity	[66]
Tilted FBG with gold coating	523.41 nm/RIU	1.15 to 1.143 RIU	Multi-angle tilted FBG with gold coating	Biochemical sensing in hard-to-reach places, in vivo measurements	Cross-sensitivity with temperature	[67]
Micro-sapphire FBG	0.9–4.2 nm/RIU	1.33 to 1.75	Point-by-point inscription followed by wet etching	High-temperature sensing (up to 1400 °C) in harsh environments	Limited by material refractive index; nonlinear sensitivity	[68]

**Table 3 sensors-25-02289-t003:** FBG for displacement monitoring in different sectors.

Application	Measurement Range (mm)	Sensitivity (pm/mm)	Environment	Benefits	Ref.
Civil Structures (Bridges, Dams)	0–50 mm	23.96	Outdoor, dynamic	High durability, real-time monitoring, resistance to EMI	[73]
Long-range Industrial Applications	0–150 mm	23.80	Harsh industrial environments (−40 to 120 °C)	High endurance, suitable for cyclic operations	[74]
Tunnel Monitoring	0–25 mm	19.48	Tunnel linings	Real-time monitoring, high sensitivity, temperature compensation	[75]
Railway Infrastructure (Track Deformation)	0–170 mm	24.8	High-speed rail systems	High precision, wide measurement range, real-time monitoring	[76]
Floating Slab Track	0–90 mm	34.32	Alternating displacement, machinery equipment	High resolution (0.0029 mm)	[77]
Structural Health Monitoring: Crack Variation in Buildings	0–110 mm	39.47	Structural components in power plants	High sensitivity, good linearity	[78]

**Table 4 sensors-25-02289-t004:** FBG for liquid level monitoring.

Application	Measurement Range	Sensitivity	Environment	Benefits	Ref.
Industrial storage tanks	0–0.25 m	27 pm/cm	Corrosive liquids	Safety in explosive environments	[79]
Water level monitoring (rivers and reservoirs)	Up to 18 m	43.5–155.7 pm/m	Reservoirs and rivers	Robust design, continuous monitoring, customizable sensitivity	[80]
Oil and gas industry	Pressure: 0–40 MPa, temperature: 25–200 °C	Pressure: 24.05 pm/MPa, temperature: 31.16 pm/°C	Harsh downhole environments (high temp, high pressure)	Small size (20 mm), high sensitivity	[81]
Industrial wastewater analysis	95.73 ppm and 0.008 ppm	0.76 pm/ppm (Cl), 38.6 pm/ppm (Pb)	Treated wastewater effluent	Accurate detection of chloride ions to prevent soil salinity.	[82]
Liquid level	Up to 10.54 m	2.74 pm/mm	Industrial tanks	High sensitivity	[83]

**Table 5 sensors-25-02289-t005:** FBG for temperature and pressure sensing.

Application	Temperature Range	Pressure Range	Temperature Sensitivity	Pressure Sensitivity	Refs.
Biomedical Monitoring	35–45 °C	102–219 mmHg	23.4 pm/°C	0.5 pm/mmHg	[88,89]
Oil and Gas	25–200 °C	0–40 MPa	31.16 pm/°C	24.05 pm/MPa	[81]

**Table 6 sensors-25-02289-t006:** FBG for strain monitoring.

Application	Strain Range (με)	Sensitivity	Environment	Benefits	Refs.
Bridges	N/A	1.15 pm/με	Dynamic bridge loads	High sensitivity, stability	[90,91]
Flexible Aerospace Parts	0–2000	~1.2 pm/με	Dynamic aerospace conditions	Lightweight, durable	[92]
High-rise Buildings	0–1000	0.5013 pm/με	High-rise construction	Reliable, low-maintenance	[93]

**Table 7 sensors-25-02289-t007:** FBG for environmental and biochemical sensing.

Application	Target Parameter	Measurement Range	Site	Benefits	Ref.
Environmental Applications					
Gas Detection	Acetylene (C_2_H_2_)	Differentiation of various C_2_H_2_ concentrations with high accuracy	Laboratory	Robust detection in mixed gas environments; minimal interference from contaminants.	[101]
Water Quality	Chlorine Concentration	1–10 ppm	Freshwater	Highly efficient sensor for detecting low chlorine levels (1–10 ppm); suitable for medical applications (e.g., dialysis).	[102]
Biochemical Applications					
Clinical Glucose Monitoring	Glucose Levels (mg/dL)	Accurate glucose measurement up to 200 mg/dL	Radial Artery (Non-invasive)	Non-invasive, highly accurate glucose monitoring using pulse wave signals measured by FBG sensors.	[103]
SARS-CoV-2 Detection	Viral Load (copies/mL)	LOD reduced by ~70% from 100.05 to 29.97 copies/mL	Laboratory Setup	Highly sensitive with reduced LOD (∼70% improvement post amplification); specific for SARS-CoV-2 strains.	[104]

**Table 8 sensors-25-02289-t008:** FBG vs. MEMS sensors—key comparative values.

Parameter	FBG Sensors	MEMS Sensors	References
Sensitivity	Strain sensitivity: 6.2 pm/με Temperature sensitivity: 50.8 pm/K Pressure sensitivity: 90.6 pm/psi	Strain sensitivity: 0.03 mV/με Temperature sensitivity: 1.12 Ω/°C Pressure sensitivity: 0.009 mV/kPa.	[54,122,123,124,125,126]
Measurement Range	Pressure range: 0 to 40 psi Temperature range: −20 to 60 °C	Strain range: ±4000 με Temperature range: −50 to 50 °C	[123,125,127,128]
Measurement Speed	Strain measurement speed: 100 kHz	From 4 s to 46 min	[129,130]
Uncertainty	−3.63 to +3.47%	0.6%	[131,132]

**Table 9 sensors-25-02289-t009:** FBG vs. Other optical fiber sensors—key comparative values.

Parameter	FBG Sensors	Interferometric Sensors (e.g., Mach–Zehnder)	Long Period Grating Sensors	Distributed Sensors	References
Temperature Sensitivity	~10–50 pm/°C	~10–1700 pm/°C	~30–100 pm/°C	0.032–10 pm/°C	[54,88,133,134,135,136,137,138,139,140,141,142,143]
Pressure Sensitivity	24–~13,100 pm/MPa	~3.7 × 10^3^–12 × 10^6^ pm/MPa	~−12–−18 nm/MPa	~0.006–0.781 pm/MPa	[81,125,144,145,146,147,148,149,150,151]
Strain Sensitivity	1.2–6.2 pm/με	~1–165 pm/με	~2–3 pm/με	~1.2 pm/με	[92,126,142,152,153,154,155,156,157,158]

**Table 10 sensors-25-02289-t010:** Overview of FBG interrogation methods and their costs.

Interrogation Method	Principle	Cost	Applications	Ref.
Spectrometer-based Interrogation	Measures the spectrum of light reflected by the FBG to detect Bragg wavelength shifts.	High cost due to the need for high-performance components, offering precise measurements.	Research, high-precision sensing, industrial applications.	[163]
Wavelength-Shift Interrogation (Fixed Filter)	Uses a fixed optical filter to detect shifts in the FBG’s reflected wavelength.	Less expensive than spectrometers but may be less accurate; limited sensor capacity.	Applications with fewer sensors or where high precision is not critical.	[164]
Optical Time Domain Reflectometry	Uses time-domain measurements to locate and interrogate FBG sensors along a fiber.	More expensive than basic wavelength-shift systems; suitable for long-distance sensing or distributed sensors.	Large networks, long-distance sensing, structural health monitoring.	[165]
Addressed FBG Sensors	Assigns a unique address to each FBG sensor, allowing multiple sensors to share a single interrogator.	Significant cost reduction for large-scale networks by using one interrogator for many sensors.	Large-scale, distributed monitoring systems, such as environmental and structural health monitoring.	[166]

## Data Availability

No new data were created or analyzed in this study.

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
