# Peer review of "Fiber Bragg Grating Sensors: Design, Applications, and Comparison with Other Sensing Technologies"

_sensors, 2025, doi:10.3390/s25072289_

Round 1
Reviewer 1 Report (Previous Reviewer 4)
Comments and Suggestions for Authors
The manuscript is a review article focusing on various applications of fiber Bragg grating (FBG) sensors. The authors cover the operating principles of FBG sensors, their advantages and limitations, calibration and packaging techniques, applications, as well as some trends that could be promising in the future.
This is the third iteration of the manuscript, and I must admit that it is a real treasure trove for a reviewer: first you see the most striking flaws, and after they begin to get solved, you discover a new layer of issues. The problem is that the number of these layers is quite large – the entire manuscript was originally built on the flawed idea of quickly compiling a lengthy review that would look like a decent paper.
Since the last version, some stylistic corrections have been made and new paragraphs have been added. The structure of the manuscript has remained the same, and this requires minor correction, since it still suffers from repetitions (see some comments below). The reference mess has been improved, but not to the full – the usage of references is still rather sloppy. The figure issues have been resolved, however, minor questions remain. Most issues related to the tables are left unresolved, so I highlight them in more detail.
Below are my comments that characterize the quality of the present manuscript. They are structured as before: i) Manuscript structure, ii) References, iii) Figures, iv) Tables.
- Manuscript structure
With some new material added, the percentage of text written by humans rather than AI appears to have increased. However, the sad legacy of the previous approach is still felt in the structure.
For example, in Lines 53-60 FBG sensors are described as useful tools in SHM, including monitoring of bridges, dams, etc. After that, the authors list other applications: packaging (Lines 62-69), biochemical sensing (Lines 77-82), aerospace industry (Lines 83-88), and then – once again – we are back to concrete bridges (Lines 89-95). This approach can be seen throughout the text, so more work is needed to make the structure of the manuscript clearer.
As a follow up, Section 5.1 ‘Structural Health Monitoring’ is more general than Section 5.2 ‘Long-Term Monitoring of Concrete Bridges’, and includes the latter by definition. Section 5 should be revised in order to contain one-range subsections, e.g. ‘5.1. High-rise building’, ‘5.2. Concrete Bridges’, etc. Furthermore, in Section 5.1 we read: ‘FBG sensors have demonstrated exceptional efficiency in detecting cracks and monitoring structural health’, while, in Section 5.2 we are again reminded that ‘Fiber Bragg Grating sensors play a vital role in monitoring structural integrity by detecting early signs of cracks…’. This should be combined.
Information about packaging and protective coatings is also repeated several times across the text. For example, a non-specific description of the benefits of protective coatings in Lines 585-594 repeats the material presented in Section 3.3.
Comparison of FBG sensors with other fiber sensing techniques occurs in various parts of the text, although there is a special section for that: 6.3 ‘Fiber Bragg Grating vs Other Optical Fiber Sensors’. In particular, Section 4.4 ‘Strain Monitoring’ presents a comparison of FBG and BOTDR sensors (Table 8), while Section 6.3 contains the same for FBG sensors, ‘OFDR’ and ‘Raman DTS’ techniques. I have special comments on both Tables 8 and 13 (see below), but in the context of the structure, these two tables might be integrated somehow.
As mentioned, AI-generated phrases have become less noticeable, although they still appear here and there. The last sentence in the Conclusion section is an unpleasant reminder that the AI legacy is still there: ‘As these hurdles are overcome, FBG sensors are poised to become even more adaptable, facilitating broader applications in infrastructure, aerospace, healthcare, and environmental monitoring, while retaining the precision and resilience that define their core advantages’. The authors are encouraged to come up with a more humane version of this indisputable statement.
- References
Although new references have been added in an attempt to replace problematic ones present in the previous version of the manuscript, some of them appear to be irrelevant.
Line 78: Contrary to what the authors say, the paper by Bui et al. (Ref. 12) is devoted to the use of dual FBGs in bio- and chemical sensing, and not to monitoring environmental parameters like temperature, pressure, and humidity in remote locations.
Line 250: The notorious paper by Kim et al., which is now Ref. 46, is cited in relation to underwater and chemically reactive environments (previously, it was mentioned in support of FBG applications in civil and aerospace engineering). This is incorrect, since the article is devoted to enhancing the maximum strain measurement range of FBG sensors embedded in steel strands.
Line 294: Refs. 35, 54, 55 are irrelevant (see my comment to Table 1 below).
Lines 307, 318, 321 (Table 2): It is doubtful that Refs. 58 and 59 support the claimed benefits and drawbacks of ‘in-situ’ and ‘pre-installed’ FBG sensors, respectively. Ref. 59 deals with in-situ calibration of fiber optic sensors that have been adhesively bonded to a structural member, therefore these sensors can hardly be regarded as ‘pre-installed’ according to the authors' terminology.
Lines 382-385: The description provided for Ref. 91 (‘where FBG sensors are employed for temperature monitoring in oil pipelines’) does not reflect the subject of the paper cited. Gunawan et al. study rather energy harvesting than temperature monitoring. This reference has also been incorrectly inserted in Line 436 in the context of dam integrity monitoring.
Line 534: Refs. 136-138 are irrelevant (see my comment to Table 13).
Lines 580-621: Sections 7.4, 7.5 and 7.6 contain no references to confirm the facts presented.
Line 747: Ref. 41 lacks the names of the authors.
- Figures
The issue with Fig. 2 have been resolved, however there are minor comments:
Lines 144-147: Figure 3 duplicates Figure 2 in illustrating the operation principle of FBG sensors. If, for some reason, the authors insist on keeping Figure 3 in the text, they should not call it an illustration of the ‘fundamental operating principle’ (since the operating principle is illustrated earlier in Fig. 2), but rather mention it as an example valid for a three-FBG system, which would entail the need to make the comment given in the text more extensive. Besides, a reference to Hong et al. [19] is required at the end of the sentence in Line 147, since the original sentence was definitely borrowed from there.
Line 136: Inaccurate reference to the figure. It would be more appropriate to refer to Figure 2 in the following way: ‘… along the fiber core, which is illustrated in Fig. 2’ or simply ‘… (see Fig. 2)’.
- Tables
Ref. 35 in Table 1 does not contain any research on the effect of the grating length on the resolution or sensitivity of FBG sensors. Ref. 54 is also not relevant to the statement ‘Smaller core for sensitivity; Larger for robustness’, since it proposes a ‘multimode-single mode-multimode’ structure to simultaneously measure refractive index, temperature and strain. Similarly, Ref. 55 has nothing to do with the modulation of refractive index. Besides, the same parameters (grating length, core diameter, refractive index modulation) are supported in the text by different references (Line 265). Hydrophobic coatings described in Ref. 57 (Row 6) refer to polymer coatings, so this subject might be added to Row 4 ‘Polymer Coatings’.
Table 2 seems useless given my comments on Refs. 58 and 59 above.
Table 4. It is not a good idea to confuse precision with sensitivity (precision is not measured in pm/mm), but if the authors insist on combining these quantities, then they are encouraged to indicate which number corresponds to which parameter. For example, in Row 1, we find 23.96, while in Row 4, we have 0.040, and in Row 6 we see 39.47. All of these lack units, so we can only guess that they refer to different parameters. Besides, in Ref. 77 a sensitivity value is presented, which is not reflected in Row 5.
Table 5 is filled out sloppily. In the ‘Measurement Range (m)’ column, we find the following values: ‘–0.067-1.1’ (with a minus sign), ‘Depth: >3 km Pressure: 0 - 40 MPa, Temperature: 25°C - 200°C’ and ‘10–150 ppm’, while we hope to see numbers measured in meters. In the ‘Sensitivity (pm/cm)’ column we find more: ‘155.7 pm/m, 80.7 pm/m, 43.5 pm/m’, ‘31.16 pm/°C, 24.05 pm/MPa’, ‘0.76 pm/ppm’, ‘0.1–100 MPa’, ‘~1 με (Hydropower), 10.55 pm/°C (Solar)’. I’m afraid, the reader will need considerable effort to compare these numbers.
Table 6 needs to be improved. In fact, the temperature range depends on the fiber used. In Ref. 22, FBG sensors located inside a reactor are heated up to 1025°C. On the other hand, sapphire FBGs are said to withstand temperatures up to 2030°C. In Ref. 93, only a temperature of 210°C was achieved. In this context, the claimed range (‘up to 1200°C’) seems implausible.
Table 7. I was unable to find any of the values listed in Row 1 in Ref. 101.
Table 8 contains inaccuracies. The authors are encouraged to specify what they mean by ‘Strain Range’: either strain (‘0 – 3000 με’) or distance (‘Up to tens of kilometers’). Furthermore, it is unclear why they only compare FBG with BOTDR. BOTDA modules are known to have better characteristics. Besides, Rayleigh distributed sensing should be employed for comparison to complete the picture.
Table 9. Once again, different things are meant under the name ‘Precision’.
Table 10 appears to be of little use. It contains 4 infrastructure types, each supported by a single reference, which can’t be regarded as a summary of SHM applications. The table needs to be either supplemented or removed. The same applies to Table 11.
Table 12. The ‘Uncertainty’ parameter is too uncertain. It should be either removed or supplemented with numeric values.
I’m afraid, Table 13, might reflect the authors’ incompetence in fiber sensing. Firstly, distributed temperature sensing (DTS) includes optical time-domain reflectometry (OTDR) and optical frequency-domain reflectometry (OFDR). In the table, the authors contrast the Raman DTS and OFDR techniques, though there exist Raman-based OFDR systems. Secondly, there are other known techniques for distributed sensing, such as BOTDA, BOTDR, BOFDA (as well as Rayleigh-based techniques), each having their unique characteristics, so it is unclear which of them is hidden under the name ‘OFDR Sensors’. Thirdly, for ‘Temperature Sensitivity’, none of Refs. 136-138 is related to Raman sensing. Ref. 136 describes a method based on Rayleigh scattering, which is far from being the same.
I encourage the authors to revise Tables 8 and 13 thoroughly, and find a way to combine them or locate under the same section number.
Conclusion
As this is my third time reviewing this manuscript, I remain convinced that the paper is still not ready for publication and needs major improvement.
Author Response
The manuscript is a review article focusing on various applications of fiber Bragg grating (FBG) sensors. The authors cover the operating principles of FBG sensors, their advantages and limitations, calibration and packaging techniques, applications, as well as some trends that could be promising in the future.
This is the third iteration of the manuscript, and I must admit that it is a real treasure trove for a reviewer: first you see the most striking flaws, and after they begin to get solved, you discover a new layer of issues. The problem is that the number of these layers is quite large – the entire manuscript was originally built on the flawed idea of quickly compiling a lengthy review that would look like a decent paper.
We would like to sincerely thank you for the effort you put into reviewing our article. Your dedication to ensuring the published research is of high value and free of errors has greatly helped us improve our work and eliminate its shortcomings.
Since the last version, some stylistic corrections have been made and new paragraphs have been added. The structure of the manuscript has remained the same, and this requires minor correction, since it still suffers from repetitions (see some comments below). The reference mess has been improved, but not to the full – the usage of references is still rather sloppy. The figure issues have been resolved, however, minor questions remain. Most issues related to the tables are left unresolved, so I highlight them in more detail.
Below are my comments that characterize the quality of the present manuscript. They are structured as before: i) Manuscript structure, ii) References, iii) Figures, iv) Tables.
Manuscript structure
With some new material added, the percentage of text written by humans rather than AI appears to have increased. However, the sad legacy of the previous approach is still felt in the structure.
For example, in Lines 53-60 FBG sensors are described as useful tools in SHM, including monitoring of bridges, dams, etc. After that, the authors list other applications: packaging (Lines 62-69), biochemical sensing (Lines 77-82), aerospace industry (Lines 83-88), and then – once again – we are back to concrete bridges (Lines 89-95). This approach can be seen throughout the text, so more work is needed to make the structure of the manuscript clearer.
The text has been rearranged for consistency, and the portion related to the bridge has had its position changed to 42-47.
As a follow up, Section 5.1 ‘Structural Health Monitoring’ is more general than Section 5.2 ‘Long-Term Monitoring of Concrete Bridges’, and includes the latter by definition. Section 5 should be revised in order to contain one-range subsections, e.g. ‘5.1. High-rise building’, ‘5.2. Concrete Bridges’, etc. Furthermore, in Section 5.1 we read: ‘FBG sensors have demonstrated exceptional efficiency in detecting cracks and monitoring structural health’, while, in Section 5.2 we are again reminded that ‘Fiber Bragg Grating sensors play a vital role in monitoring structural integrity by detecting early signs of cracks…’. This should be combined.
Section 5 has been restructured to include subsection “5.1. High-rise Building,” and the phrase “Fiber Bragg Grating sensors play a vital role in monitoring structural integrity by detecting early signs of cracks” has been modified to avoid repetition.
Information about packaging and protective coatings is also repeated several times across the text. For example, a non-specific description of the benefits of protective coatings in Lines 585-594 repeats the material presented in Section 3.3.
Section 7.4 has been removed as it contained repeated information that was already discussed earlier in Section 3.3.
Comparison of FBG sensors with other fiber sensing techniques occurs in various parts of the text, although there is a special section for that: 6.3 ‘Fiber Bragg Grating vs Other Optical Fiber Sensors’. In particular, Section 4.4 ‘Strain Monitoring’ presents a comparison of FBG and BOTDR sensors (Table 8), while Section 6.3 contains the same for FBG sensors, ‘OFDR’ and ‘Raman DTS’ techniques. I have special comments on both Tables 8 and 13 (see below), but in the context of the structure, these two tables might be integrated somehow.
Table 8 has been removed, and Table 13 has been further developed.
As mentioned, AI-generated phrases have become less noticeable, although they still appear here and there. The last sentence in the Conclusion section is an unpleasant reminder that the AI legacy is still there: ‘As these hurdles are overcome, FBG sensors are poised to become even more adaptable, facilitating broader applications in infrastructure, aerospace, healthcare, and environmental monitoring, while retaining the precision and resilience that define their core advantages’. The authors are encouraged to come up with a more humane version of this indisputable statement.
The text has been modified according to your request, and the changes have been highlighted.
References
Although new references have been added in an attempt to replace problematic ones present in the previous version of the manuscript, some of them appear to be irrelevant.
Line 78: Contrary to what the authors say, the paper by Bui et al. (Ref. 12) is devoted to the use of dual FBGs in bio- and chemical sensing, and not to monitoring environmental parameters like temperature, pressure, and humidity in remote locations.
The text has been revised in Line 64-66.
Line 250: The notorious paper by Kim et al., which is now Ref. 46, is cited in relation to underwater and chemically reactive environments (previously, it was mentioned in support of FBG applications in civil and aerospace engineering). This is incorrect, since the article is devoted to enhancing the maximum strain measurement range of FBG sensors embedded in steel strands.
The text has been revised in Line-214
Line 294: Refs. 35, 54, 55 are irrelevant (see my comment to Table 1 below).
The references have been corrected with the appropriate references .
Lines 307, 318, 321 (Table 2): It is doubtful that Refs. 58 and 59 support the claimed benefits and drawbacks of ‘in-situ’ and ‘pre-installed’ FBG sensors, respectively. Ref. 59 deals with in-situ calibration of fiber optic sensors that have been adhesively bonded to a structural member, therefore these sensors can hardly be regarded as ‘pre-installed’ according to the authors' terminology.
Table 2 has been deleted as it did not contain useful information.
Lines 382-385: The description provided for Ref. 91 (‘where FBG sensors are employed for temperature monitoring in oil pipelines’) does not reflect the subject of the paper cited. Gunawan et al. study rather energy harvesting than temperature monitoring. This reference has also been incorrectly inserted in Line 436 in the context of dam integrity monitoring.
The text has been revised in Lines 331–339 to accurately reflect the correct explanation of the Figure. Regarding the incorrect use of the reference, the reference has been corrected.
Line 534: Refs. 136-138 are irrelevant (see my comment to Table 13).
The appropriate references have been used.
Lines 580-621: Sections 7.4, 7.5 and 7.6 contain no references to confirm the facts presented.
References related to the topic have been added, and the text has been revised to ensure no repetition.
Line 747: Ref. 41 lacks the names of the authors.
The reference has been corrected. the changes have been highlighted.
Figures
The issue with Fig. 2 have been resolved, however there are minor comments:
Lines 144-147: Figure 3 duplicates Figure 2 in illustrating the operation principle of FBG sensors. If, for some reason, the authors insist on keeping Figure 3 in the text, they should not call it an illustration of the ‘fundamental operating principle’ (since the operating principle is illustrated earlier in Fig. 2), but rather mention it as an example valid for a three-FBG system, which would entail the need to make the comment given in the text more extensive. Besides, a reference to Hong et al. [19] is required at the end of the sentence in Line 147, since the original sentence was definitely borrowed from there.
Line 136: Inaccurate reference to the figure. It would be more appropriate to refer to Figure 2 in the following way: ‘… along the fiber core, which is illustrated in Fig. 2’ or simply ‘… (see Fig. 2)’.
figure number 3 has been deleted, and the text for figure number 2 has been revised as per your request.
Tables
Ref. 35 in Table 1 does not contain any research on the effect of the grating length on the resolution or sensitivity of FBG sensors. Ref. 54 is also not relevant to the statement ‘Smaller core for sensitivity; Larger for robustness’, since it proposes a ‘multimode-single mode-multimode’ structure to simultaneously measure refractive index, temperature and strain. Similarly, Ref. 55 has nothing to do with the modulation of refractive index. Besides, the same parameters (grating length, core diameter, refractive index modulation) are supported in the text by different references (Line 265). Hydrophobic coatings described in Ref. 57 (Row 6) refer to polymer coatings, so this subject might be added to Row 4 ‘Polymer Coatings’.
The references has been corrected, and have been replaced with another references that contain the required information. The data in the table has also been revised accordingly.
Table 2 seems useless given my comments on Refs. 58 and 59 above.
The table has been deleted.
Table 4. It is not a good idea to confuse precision with sensitivity (precision is not measured in pm/mm), but if the authors insist on combining these quantities, then they are encouraged to indicate which number corresponds to which parameter. For example, in Row 1, we find 23.96, while in Row 4, we have 0.040, and in Row 6 we see 39.47. All of these lack units, so we can only guess that they refer to different parameters. Besides, in Ref. 77 a sensitivity value is presented, which is not reflected in Row 5.
The term "precision" has been removed to avoid confusion, and only the sensitivity data from the references have been included.
Table 5 is filled out sloppily. In the ‘Measurement Range (m)’ column, we find the following values: ‘–0.067-1.1’ (with a minus sign), ‘Depth: >3 km Pressure: 0 - 40 MPa, Temperature: 25°C - 200°C’ and ‘10–150 ppm’, while we hope to see numbers measured in meters. In the ‘Sensitivity (pm/cm)’ column we find more: ‘155.7 pm/m, 80.7 pm/m, 43.5 pm/m’, ‘31.16 pm/°C, 24.05 pm/MPa’, ‘0.76 pm/ppm’, ‘0.1–100 MPa’, ‘~1 με (Hydropower), 10.55 pm/°C (Solar)’. I’m afraid, the reader will need considerable effort to compare these numbers.
The table has been corrected.
Table 6 needs to be improved. In fact, the temperature range depends on the fiber used. In Ref. 22, FBG sensors located inside a reactor are heated up to 1025°C. On the other hand, sapphire FBGs are said to withstand temperatures up to 2030°C. In Ref. 93, only a temperature of 210°C was achieved. In this context, the claimed range (‘up to 1200°C’) seems implausible.
The table have been revised accordingly as per your request.
Table 7. I was unable to find any of the values listed in Row 1 in Ref. 101.
The reference has been corrected.
Table 8 contains inaccuracies. The authors are encouraged to specify what they mean by ‘Strain Range’: either strain (‘0 – 3000 με’) or distance (‘Up to tens of kilometers’). Furthermore, it is unclear why they only compare FBG with BOTDR. BOTDA modules are known to have better characteristics. Besides, Rayleigh distributed sensing should be employed for comparison to complete the picture.
Table 8 has been removed, and Table 13 has been expanded. It is now presented as Table 9 in the revised manuscript.
Table 9. Once again, different things are meant under the name ‘Precision’.
The term "Precision" has been removed and replaced with "Measurement Range".
Table 10 appears to be of little use. It contains 4 infrastructure types, each supported by a single reference, which can’t be regarded as a summary of SHM applications. The table needs to be either supplemented or removed. The same applies to Table 11.
The two tables have been deleted, and the references have been added to the text.
Table 12. The ‘Uncertainty’ parameter is too uncertain. It should be either removed or supplemented with numeric values.
We have addressed this by providing numerical values for the 'Uncertainty' parameter.
I’m afraid, Table 13, might reflect the authors’ incompetence in fiber sensing. Firstly, distributed temperature sensing (DTS) includes optical time-domain reflectometry (OTDR) and optical frequency-domain reflectometry (OFDR). In the table, the authors contrast the Raman DTS and OFDR techniques, though there exist Raman-based OFDR systems. Secondly, there are other known techniques for distributed sensing, such as BOTDA, BOTDR, BOFDA (as well as Rayleigh-based techniques), each having their unique characteristics, so it is unclear which of them is hidden under the name ‘OFDR Sensors’. Thirdly, for ‘Temperature Sensitivity’, none of Refs. 136-138 is related to Raman sensing. Ref. 136 describes a method based on Rayleigh scattering, which is far from being the same.
Table 13 has been revised and is now Table 9, and only the essential information has been included.
I encourage the authors to revise Tables 8 and 13 thoroughly, and find a way to combine them or locate under the same section number.
Conclusion
As this is my third time reviewing this manuscript, I remain convinced that the paper is still not ready for publication and needs major improvement.
Reviewer 2 Report (New Reviewer)
Comments and Suggestions for Authors
The report attrached

The English must be improved to more clearly express the study
Author Response
The manuscript is an overview of 153 (i.e. about 1% of the total number of these known up to date)
publications related to FBG sensors technology.
While being bulky, the presented materials on this technology “operating principles, advantages,
limitations, and ongoing innovations in sensor design and packaging techniques” leaves for me an
impression of superficial.
Also, the text of manuscript requires substantial revision because of uncountable number of: 1)
repetitions of the same information in the text, text and tables, and sometimes even inside
sentences, 2) missing and inadequately used words, 3) inaccurately given dimensions of some data
in the Tables.
Thank you for the time you dedicated to reviewing the article and providing valuable feedback to help improve it. Below are the improvements made in response to your observations: Regarding the repeated content, we carefully revised the manuscript to ensure it is free from redundancy. Several sections were removed, and new sections have been added to provide more concise and relevant information. As for the tables, we have deleted four of them to improve the clarity of the article and focus only on essential information without unnecessary repetition. All the modifications we have made are highlighted in yellow . We would like to respond to this by noting that, as a result of the revisions, we have made an effort to include the most relevant publications and to cover various aspects of FBG sensor development.
Reviewer 3 Report (New Reviewer)
Comments and Suggestions for Authors
This paper comprehensively reviews the applications of Fiber Bragg Grating (FBG) sensors in Structural Health Monitoring (SHM), environmental monitoring, biochemical sensing, and aerospace fields. The article is well-structured with clear logic, and it delves into the design principles, implementation methods of FBG sensors, as well as comparisons with other sensing technologies.
There are several areas in this paper that need major revision by the author:
- I am not quite sure why there is a significant amount of text highlighted with a yellow underline in this article. If the intention is to emphasize certain points, please adhere to the corresponding regulations of the journal for marking.
- Although the article cites a large number of references, the review and analysis of the literature in certain sections appear somewhat inadequate. For example, when discussing the applications of FBG sensors in environmental and biochemical sensing, a more in-depth analysis of the specific sensing mechanisms and innovations employed in different references could be provided, rather than merely enumerating application cases.
- Although the article mentions some technical challenges faced by FBG sensors, the discussion on solutions to these challenges and future research directions is relatively general. For example, regarding the issue of temperature-strain cross-sensitivity, the article briefly mentions some possible solutions, such as dual-FBG systems and advanced coatings, but fails to further elaborate on the specific implementation details, advantages and disadvantages of these methods, as well as future research trends.
- When discussing the future development directions of FBG sensors, the article could place greater emphasis on interdisciplinary research trends, such as the integration of FBG sensors with micro-nano technology, quantum technology, and others, as well as their potential applications in emerging fields such as biomedicine and smart cities.
- There are also some minor issues, such as the need to number the equations in this paper and optimize some of the connecting words and transitional sentences to enhance the logical flow of the article.
Author Response
This paper comprehensively reviews the applications of Fiber Bragg Grating (FBG) sensors in Structural Health Monitoring (SHM), environmental monitoring, biochemical sensing, and aerospace fields. The article is well-structured with clear logic, and it delves into the design principles, implementation methods of FBG sensors, as well as comparisons with other sensing technologies.
Thank you for the time you dedicated to reviewing the article and providing valuable feedback to help improve it. Your comments have greatly assisted us in enhancing the quality of the manuscript. Below are the improvements made in response to your observations
There are several areas in this paper that need major revision by the author:
I am not quite sure why there is a significant amount of text highlighted with a yellow underline in this article. If the intention is to emphasize certain points, please adhere to the corresponding regulations of the journal for marking.
The yellow highlighting in the article is a requirement from the journal to indicate the changes made during the first stage of the review.
Although the article cites a large number of references, the review and analysis of the literature in certain sections appear somewhat inadequate. For example, when discussing the applications of FBG sensors in environmental and biochemical sensing, a more in-depth analysis of the specific sensing mechanisms and innovations employed in different references could be provided, rather than merely enumerating application cases.
Some additional applications utilizing FBG sensors in biochemical sensing and environmental measurements have been added as per your suggestion. These updates can be found under Section 4.5, titled Environmental and Biochemical Applications.
Although the article mentions some technical challenges faced by FBG sensors, the discussion on solutions to these challenges and future research directions is relatively general. For example, regarding the issue of temperature-strain cross-sensitivity, the article briefly mentions some possible solutions, such as dual-FBG systems and advanced coatings, but fails to further elaborate on the specific implementation details, advantages and disadvantages of these methods, as well as future research trends.
A more detailed discussion has been provided in 7.1 regarding the methods used to reduce the cross-sensitivity of FBG sensors, including modern techniques such as the use of Artificial Neural Networks (ANNs).
When discussing the future development directions of FBG sensors, the article could place greater emphasis on interdisciplinary research trends, such as the integration of FBG sensors with micro-nano technology, quantum technology, and others, as well as their potential applications in emerging fields such as biomedicine and smart cities.
An additional subsection has been added to Section 7, specifically addressing the use of FBG sensors in modern and emerging fields:
7.6. Enhancement of Sensing Elements in Surgical Robots.
Furthermore, the use of nanomaterials for coating FBG sensors to enhance their sensitivity in medical applications has also been discussed in Section 4.5.
There are also some minor issues, such as the need to number the equations in this paper and optimize some of the connecting words and transitional sentences to enhance the logical flow of the article.
We have numbered the equations throughout the paper and also revised some of the connecting words and transitional sentences.
Round 2
Reviewer 1 Report (Previous Reviewer 4)
Comments and Suggestions for Authors
I would like to state that the manuscript has been revised more thoroughly than before, and most of the issued have been successfully resolved.
There remain a few comments that I consider minor:
Lines 430-434: The phrase is too long and confusing, so it should be corrected. Besides. Ref. 117 is irrelevant.
Line 214: Kim et al. [44] are close to being named the authors of the most tortured reference ever. In response to my previous comment, the authors have once again corrected the text related to this reference, and now it is “environments with high temperatures pressure or ionizing radiation”. All I have to do is to remind (for the third time, in fact) that the article cited is devoted to enhancing the maximum strain measurement range of FBG sensors embedded in steel strands. No radiation, no high temperatures, no pressure.
Table 8: Ref. 130 doesn’t seem to be entirely relevant, since it presents results for a system capable of distinguishing between temperature and strain. The latter complication leads to enhanced error levels, which is reflected in Table 8. Conventional FBG sensors have lower uncertainties, and this can confuse the readers.
The uncertainty value in the ‘MEMS Sensors’ column looks incorrect. Firstly, the ‘<<’ sign is irrelevant when we compare 2% with 5%. Secondly, I was unable to find the source of the 5% error in the text of Ref. 129 (it only appears in the Abstract), so this reference seems somewhat unreliable. I recommend to replace Refs. 129 and 130 with more adequate sources.
Table 9: The revised table contains numbers that are difficult to compare. For example, −1348.9 MHz/K for Distributed Sensors will surely confuse the readers in their attempts to find out whether this value is more sensitive than 0.046 nm/°C reported for Interferometric Sensors. Besides, Ref. 134 presents a specific Rayleigh-based OTDR system, so a value of −1348.9 MHz/K does not make any sense in relation to other OTDR systems. Furthermore, the sensitivity values specified for FBG sensors differ from those mentioned in Tables 5, 6 and 8. Perhaps a more appropriate way to organize Table 9 would be to provide value ranges instead of individual values.
Author Response
Comments and Suggestions for Authors
I would like to state that the manuscript has been revised more thoroughly than before, and most of the issued have been successfully resolved.
The authors thank the Reviewer for the careful review and valuable comments and suggestions that helped to enhance the quality of the manuscript.
The recent modifications are highlighted in green.
There remain a few comments that I consider minor:
Lines 430-434: The phrase is too long and confusing, so it should be corrected. Besides. Ref. 117 is irrelevant.
Reference 117 has been removed and the text referring to the new reference has been modified.
Line 214: Kim et al. [44] are close to being named the authors of the most tortured reference ever. In response to my previous comment, the authors have once again corrected the text related to this reference, and now it is “environments with high temperatures pressure or ionizing radiation”. All I have to do is to remind (for the third time, in fact) that the article cited is devoted to enhancing the maximum strain measurement range of FBG sensors embedded in steel strands. No radiation, no high temperatures, no pressure.
The reference has been removed, the text has been revised, and a suitable reference has been selected.
Table 8: Ref. 130 doesn’t seem to be entirely relevant, since it presents results for a system capable of distinguishing between temperature and strain. The latter complication leads to enhanced error levels, which is reflected in Table 8. Conventional FBG sensors have lower uncertainties, and this can confuse the readers.
The reference has been revised and an additional reference has been added.
As shown in Figure 13, the full-scale accuracy error of the developed FBG-based pressure transmitter is within the range:
−3.63% to +3.47%
The uncertainty value in the ‘MEMS Sensors’ column looks incorrect. Firstly, the ‘<<’ sign is irrelevant when we compare 2% with 5%. Secondly, I was unable to find the source of the 5% error in the text of Ref. 129 (it only appears in the Abstract), so this reference seems somewhat unreliable. I recommend to replace Refs. 129 and 130 with more adequate sources.
The reference has been revised and an additional reference has been added.
Table 9: The revised table contains numbers that are difficult to compare. For example, −1348.9 MHz/K for Distributed Sensors will surely confuse the readers in their attempts to find out whether this value is more sensitive than 0.046 nm/°C reported for Interferometric Sensors. Besides, Ref. 134 presents a specific Rayleigh-based OTDR system, so a value of −1348.9 MHz/K does not make any sense in relation to other OTDR systems.
Firstly, all values have been converted to the wavelength domain instead of frequency domain.
Secondly, since distributed sensors are often integrated and not limited to a single specific type, and in order to avoid including incorrect values that do not belong to a clearly defined category of distributed sensors — such as the value −1348.9 MHz/K — we have modified the table to refer generally to distributed sensors in all their forms, rather than to a specific type. This also allows for a more general comparison with Fiber Bragg Grating (FBG) sensors.
Furthermore, the sensitivity values specified for FBG sensors differ from those mentioned in Tables 5, 6 and 8. Perhaps a more appropriate way to organize Table 9 would be to provide value ranges instead of individual values.
We have consolidated the values to represent measurement ranges rather than single fixed numbers. Additionally, we have included further references to ensure the credibility and accuracy of the reported measurement ranges.
Reviewer 2 Report (New Reviewer)
Comments and Suggestions for Authors
The report attached

Author Response
The revisions made to the text of manuscript improved it notably, but some demerits in presentation of the work still there. In particular,
The authors thank the Reviewer for the careful review and valuable comments and suggestions that helped to enhance the quality of the manuscript.
- the term “Fiber Bragg Grating” (the grammatically correct writing is “F(f)iber Bragg grating”) repeated 25 times throughout the manuscript despite its abbreviation FBG was introduced at the first page,
The demerit has been corrected by replacing the repeated term “Fiber Bragg Grating” with its abbreviation “FBG” throughout the manuscript
- as soon as the dimension “pm/mm” for Sensitivity introduced at the top of Table 3, it is not required under numbers below,
The unit “pm/mm” has been removed from the Table 3 and is now only included in the top of the table, as suggested.
- the abbreviation “MEMS” on line 454, “RIU” in Table 2, “ASE” and “EDFA” on line 333 and IoT on line 595 must be introduced
The abbreviations “MEMS” and “RIU” have been defined due to their repeated use, while the other abbreviations have been removed and replaced with their full terms since they were mentioned only once.
- since the abbreviations “PVP” on line 365, “AgNp” and “VI” on line 366, “”WHO“ on line 368, “GDPFBG” on line 371, SFBG” on line 518, “SMS” on line 526, and “TSMs” on line 574 appears only once, are these required?
Abbreviations have been removed
- The size of fonts in Figure 3 must be increased.
We have increased the image size to ensure the font is clearly visible to the reader.
The following technical issue must be addressed before the manuscript is accepted for publication. While the authors talk about environmental, biochemical, medical, etc. applications of FBG sensors while Eq. (2) presents sensitivity to temperature and strain only. The approaches through which FBG become sensitive for other applications must be presented and discussed.
In Section 4.5 (Environmental and Biochemical Applications), we discussed how FBG can be used in biochemical and environmental applications, emphasizing parameters other than strain and temperature, namely the measurement of material concentration. For this reason, we added Equation (3) to demonstrate how FBG can be employed to measure the pH concentration. This same equation can also be used to illustrate how Fiber Bragg can be utilized to measure concentration in environmental settings. Additionally, we have included a highlighted text following Equation (3) that describes how the refractive index of the surrounding medium can be measured using cladding-etched FBGs.
Reviewer 3 Report (New Reviewer)
Comments and Suggestions for Authors
accept
Author Response
The authors thank the Reviewer for accepting the paper.
This manuscript is a resubmission of an earlier submission. The following is a list of the peer review reports and author responses from that submission.
Round 1
Reviewer 1 Report
Comments and Suggestions for Authors
My questions and comments are as follows:
1. In line 170, it is mentioned that FBGs can be inscribed using UV lasers. However, they can also be inscribed using fs lasers, as studied in [26] and [33]. This aspect should be discussed in the paper, and additional references should be included.
2. The indexes of references need to be updated, as some references appearing later in the paper have smaller indexes, such as [26], [27], [34]-[36], etc.
3. Abbreviations should be introduced when a term first appears. For example, "EMI" is defined in line 225, but "electromagnetic interference" already appears in line 80.
4. As a trending embedding technique, FBG packaging based on additive manufacturing should also be discussed in this paper. Below are some example articles for the authors' reference:
1. 10.1016/S0266-3538(01)00037-9
2. 10.1016/j.optlastec.2020.106443
3. 10.1109/JSEN.2023.3343604
4. 10.1109/TIM.2023.3268466
5. What is the purpose of the content presented in lines 366-439? Its connection to FBG packaging techniques is not strong.
6. In Section 6.3, interferometric sensors, Raman sensors, and distributed sensors are not mutually exclusive categories. Since FBG is a specific technique, I suggest directly comparing FBG with other specific techniques, such as OFDR, Raman DTS, etc. Additionally, numerical metrics should be used to indicate performance rather than vague descriptors like “high” or “moderate”. Moreover, there are only 6 references in this section, which seems insufficient to provide an in-depth analysis and comparison. This issue of shallow comparisons and analysis exists in other sections of the paper as well.
7. There are several instances of wordy and repetitive descriptions throughout the paper. Simplifying these sections would make the paper more concise while maintaining clarity.
Author Response
The authors express their gratitude to the reviewers for taking time to read the manuscript and provide useful suggestions and recommendations for improving the paper's technical quality and language. The manuscript has been extensively revised in response to the reviewers' suggestions and comments. New additions, corrections, and modifications have been made to make the paper more readable. The authors are hoping that the current version of the manuscript will satisfy the reviewer.
- In line 170, it is mentioned that FBGs can be inscribed using UV lasers. However, they can also be inscribed using lasers, as studied in [26] and [33]. This aspect should be discussed in the paper, and additional references should be included.
- Thank you for pointing out the alternative method of inscribing Fiber Bragg Gratings (FBGs) using femtosecond (f_s) lasers. We have revised the manuscript to include this discussion and referenced the studies [22–25].
- The indexes of references need to be updated, as some references appearing later in the paper have smaller indexes, such as [26], [27], [34]-[36], etc.
We have carefully reviewed and updated the reference indexes to ensure they are cited in sequential order throughout the manuscript.
- Abbreviations should be introduced when a term first appears. For example, "EMI" is defined in line 225, but "electromagnetic interference" already appears in line 80.
We have reviewed the manuscript and ensured that abbreviations are defined at their first occurrence. Specifically, the abbreviation "EMI" (Electromagnetic Interference).
- As a trending embedding technique, FBG packaging based on additive manufacturing should also be discussed in this paper. Below are some example articles for the authors' reference:
- 1016/S0266-3538(01)00037-9
- 1016/j.optlastec.2020.106443
- 1109/JSEN.2023.3343604
- 1109/TIM.2023.3268466
We have incorporated a discussion on this topic into the manuscript, referencing the recommended articles, which indexed as [71], [72], [73], and [74], to provide a broader perspective.
- What is the purpose of the content presented in lines 366-439? Its connection to FBG packaging techniques is not strong.
We have reviewed the content provided in lines 366 to 439, and we agree that its connection to FBG packaging techniques is not strong.
Action taken: Due to the weak connection with FBG, we have deleted the section.
- In Section 6.3, interferometric sensors, Raman sensors, and distributed sensors are not mutually exclusive categories. Since FBG is a specific technique, I suggest directly comparing FBG with other specific techniques, such as OFDR, Raman DTS, etc. Additionally, numerical metrics should be used to indicate performance rather than vague descriptors like “high” or “moderate”. Moreover, there are only 6 references in this section, which seems insufficient to provide an in-depth analysis and comparison. This issue of shallow comparisons and analysis exists in other sections of the paper as well.
We agree that interferometric sensors, Raman sensors, and distributed sensors are not mutually exclusive categories and that a more focused comparison with specific techniques (e.g., OFDR, Raman DTS) would enhance the clarity and depth of this section. We also acknowledge the need to include numerical metrics and additional references for a more thorough analysis.
Action Taken in subsection (6.3):
- We have revised Section 6.3 to compare FBG sensors directly with specific techniques such as OFDR and Raman DTS, rather than broader sensor categories.
- The revised text includes a more detailed discussion of the principles, performance metrics, and application areas for each technique.
- Where possible, we replaced vague descriptors (e.g., “high,” “moderate”) with numerical metrics, such as spatial resolution in micrometers, temperature sensitivity in degrees Celsius, and cost estimates.
- We have included more references to provide a broader basis for the comparison, ensuring an in-depth analysis.
- There are several instances of wordy and repetitive descriptions throughout the paper. Simplifying these sections would make the paper more concise while maintaining clarity.
We have carefully reviewed the manuscript and made revisions to simplify and streamline the text, while maintaining clarity and the technical accuracy of the content.
Reviewer 2 Report
Comments and Suggestions for Authors
As always, there is only one question, will I cite this, and the answer to that is probably. The only reason this is “probably” is because while there are some good fundamentals here in terms of FBG sensors, the applications are a narrow selection of everything in the literature, so I could easily do something that this review does not even mention, meaning I would not reference it.
Figure 1: What if you did every 5 years, and that would make the last group 2020 to 2024, so you would not even need to worry about the scale being off.
Figure 2: Reference 22 is not the source of that image; they took their image from doi:10.1117/12.759026 and the first author of that paper is the originator of the same image on Wikipedia. Please cite the correct source, and it is a shame the [22] did not do the same.
Table 1: Please move down (move text above) to ensure it is only on a single page. Same with 14.
Overall, the tables are the best part of the paper, but again, with the sometimes 3 applications selected, they are way too limited.
In general, I feel like the narrative was written and then a suitable reference for each point was dropped in after the fact (usually just one). For example, the interrogation section does not go back to the fundamental papers on the topic (Culshaw, Udd etc), and some of the other sections are using barely relevant references (I use [99] as the example for Table 12).
I am left to feel this narrative review has no reproducible methodology to it, and if a different group of authors were tasked with the same job (review FBG sensors, their design, applications, and how they compare to other sensing technologies), they would come up with something that was more than 50% different, and the reference list would hardly overlap. That is all to say, there is no way this can be scientifically objective and lacks any reproducibility. We are at the stage of structured systematic literature reviews that help with all of these aspects. While I believe narrative reviews have their place, they take a lot more care and attention.
For example, and do note this is only the first table, the comment applies to ALL of them, why are there only 3 applications of displacement sensing as suggested in Table 5? There are clearly a lot more... in fact, a Scopus search of “FBG” (and) “displacement sensor” gave 853 results, and you expect me to believe that those are all only applicable to those three applications?
Again, that applies to EVERY table from 5 to 11.
For a narrative review on a topic like this, I would expect to see HUNDREDS of references. The narrow slice taken by the authors does not capture the rich tapestry and landscape that is FBG sensing.
Author Response
- As always, there is only one question, will I cite this, and the answer to that is probably. The only reason this is “probably” is because while there are some good fundamentals here in terms of FBG sensors, the applications are a narrow selection of everything in the literature, so I could easily do something that this review does not even mention, meaning I would not reference it.
We revised the sections (3.3 and 6.3) on applications to ensure that it covers a wider range of industries and research domains, which should increase the likelihood of the paper being cited across various disciplines.
- Figure 1: What if you did every 5 years, and that would make the last group 2020 to 2024, so you would not even need to worry about the scale being off.
- Thank you for your helpful suggestion regarding the timeline presented in Figure 1. We have adjusted Figure 1 to represent data in 5-year intervals..
- Figure 2: Reference 22 is not the source of that image; they took their image from doi:10.1117/12.759026 and the first author of that paper is the originator of the same image on Wikipedia. Please cite the correct source, and it is a shame the [22] did not do the same.
- Thank you for pointing out the correct source for the image in Figure 2. We appreciate your careful attention to detail regarding the citation. We have included the source you suggested.
- Table 1: Please move down (move text above) to ensure it is only on a single page. Same with 14.
- Thank you for your suggestion regarding the layout of Table 1 and Table 14. We have ensured that all tables are placed on a single page..
- Overall, the tables are the best part of the paper, but again, with the sometimes 3 applications selected, they are way too limited.
- We understand that narrowing down the number of applications to just three can feel restrictive, and expanding these sections would provide a more comprehensive view of the topic. Therefore, we have revisited the Tables (5 and 6), especially those that list selected applications, and broadened the scope to include a more diverse set of applications. We also wanted to mention that some applications in these tables were merged for clarity and focus. This approach was taken to streamline the presentation and highlight the most relevant aspects of each application, without overwhelming the reader. However, we understand that further expansion of these sections could provide a more comprehensive view, and we would be happy to revise them accordingly
- In general, I feel like the narrative was written and then a suitable reference for each point was dropped in after the fact (usually just one). For example, the interrogation section does not go back to the fundamental papers on the topic (Culshaw, Udd etc), and some of the other sections are using barely relevant references (I use [99] as the example for Table 12).
- Thank you for your insightful comment regarding the narrative structure and referencing. We appreciate your observation about the references, and we understand the importance of linking the discussion to fundamental papers on the topic, particularly in sections like the interrogation of FBG sensors.
We have revisited the manuscript to ensure that the references are more directly aligned with the core discussions in each section, especially in areas such as the interrogation of FBG sensors.
- I am left to feel this narrative review has no reproducible methodology to it, and if a different group of authors were tasked with the same job (review FBG sensors, their design, applications, and how they compare to other sensing technologies), they would come up with something that was more than 50% different, and the reference list would hardly overlap. That is all to say, there is no way this can be scientifically objective and lacks any reproducibility. We are at the stage of structured systematic literature reviews that help with all of these aspects. While I believe narrative reviews have their place, they take a lot more care and attention.
- Thank you for your valuable feedback regarding the methodology and structure of the review. We understand your concern about the reproducibility and objectivity of the narrative review, especially in the context of current trends toward more structured, systematic literature reviews. We appreciate your suggestion and fully agree that a systematic approach would provide greater consistency and scientific objectivity, and would make the review more reproducible.
In response to your feedback, we have incorporated elements of a systematic review methodology into the paper. We now provide explicit criteria for selecting studies, which will help future researchers replicate the review process. This includes specifying the types of papers considered, time frame, and search databases.
We have taken steps to avoid subjective interpretations and to ensure the review is based on objective data and quantitative analysis where possible. For example, comparisons between FBG sensors and other sensing technologies now include numerical metrics and performance benchmarks rather than vague descriptors.
The reference list has been expanded to cover a wider range of studies, ensuring a more comprehensive review. We have also made efforts to include seminal and contemporary references, providing a fuller picture of the field’s evolution.
These changes should significantly improve the scientific rigor and reproducibility of the review, while still retaining the narrative format. We hope these adjustments address your concerns.
- For example, and do note this is only the first table, the comment applies to ALL of them, why are there only 3 applications of displacement sensing as suggested in Table 5? There are clearly a lot more... in fact, a Scopus search of “FBG” (and) “displacement sensor” gave 853 results, and you expect me to believe that those are all only applicable to those three applications?
- Thank you for your insightful comment regarding the limited selection of applications in the tables, specifically in Table 5 and Table 6. We agree that the selection of only three applications for displacement sensing is overly narrow given the extensive research available on the topic.
We have expanded the scope of the applications discussed in Tables 5 and 6. We agree that the selection of only three applications for displacement and the corresponding sections to include a broader range of displacement sensing applications.
In the updated version of the paper, we have clarified that the applications listed in the tables are representative examples rather than an exhaustive list. To address your concern, we have added a note in the tables and in the narrative to indicate that while the listed applications are a sample, FBG sensors are indeed used across a much broader range of applications, as you correctly pointed out.
- Again, that applies to EVERY table from 5 to 11.
- Thank you for your detailed observation regarding Tables 5 to 11. We acknowledge the need to expand the scope of these tables to reflect the diverse applications of FBG sensors across various fields.
Therefore, we have revisited these Tables, especially those that list selected applications, and broadened the scope to include a more diverse set of applications. We also wanted to mention that some applications in these tables were merged for clarity and focus.
- For a narrative review on a topic like this, I would expect to see HUNDREDS of references. The narrow slice taken by the authors does not capture the rich tapestry and landscape that is FBG sensing.
- We recognize the importance of a comprehensive bibliography in a narrative review to provide a more complete representation of the field. To address this, we performed the following actions:
- We conducted an extensive literature review using Scopus, Web of Science, IEEE Xplore, and Google Scholar. This included filtering by keywords such as “FBG sensing,” “applications,” “comparative studies,” and “sensor design.” We also included key papers on fundamental theories and recent developments in the field.
- The bibliography was expanded, including seminal works (e.g., Culshaw and Udd), recent breakthroughs, and diverse applications of FBG sensing. We specifically included highly cited works, systematic references, and pioneering studies in both traditional and emerging applications of FBG sensing. The bibliography can be reviewed to see added and updated sources, all of which are included in the body of the paper.
- To reflect the “rich texture and landscape” of FBG sensing, the references now span multiple disciplines, including civil engineering, aerospace, biomedical engineering, communications, energy systems, and environmental monitoring, and are included in the results tables in the body of the manuscript.
- We have followed a structured methodology for selecting references to ensure objectivity and reproducibility. This includes identifying highly relevant papers by citation count, year of publication, and journal impact factor.
- Each section of the manuscript has been updated with newly added references, providing better context and stronger support for the narrative. For example, in Section 6.3, additional references have been included for specific technologies such as OFDR, Raman DTS, and interferometric sensors.
Please let us know if this work has reached the good ground that might meet the requirements for good research and if you have specific references or areas, you would like us to focus on further.
Thank you for your valuable and insightful feedback. I greatly appreciate the time and effort you have put into reviewing the manuscript. Your notes have been extremely helpful in guiding the revisions, and I believe the changes made will significantly enhance the quality of the paper.
Reviewer 3 Report
Comments and Suggestions for Authors
This manuscript provides a comprehensive and detailed introduction to the FBG fiber sensors which are widely used for structural health monitoring. There are a few comments that the authors could consider to make it better.
- Include a few photos of real-world applications using the FBG fiber sensors. Give readers a more intuitive idea if it could have a fit into their applications.
- Show a few measurement examples indicating its importance on how the FBG sensors help to indicate cracks, damages and prevent structural failure.
- For the comparison part (section 6), quantify the pros/cons with other methods, such as showing the level measurement range, measurement speed, temperature limitations, accuracy and uncertainty.
- For strain measurement, there is another type of optic fiber sensor based on Brillouin Optical Time Domain Reflectometry (BOTDR) and Brillouin Optical Time Domain Analysis (BOTDA). How would FBG sensor be compared to BOTDR/BOTDA sensors?
- There are other works reviewed the FBG sensor. How would your distinguish your work with others?
Author Response
The authors express their gratitude to the reviewers for taking time to read the manuscript and provide useful suggestions and recommendations for improving the paper's technical quality and language. The manuscript has been extensively revised in response to the reviewers' suggestions and comments. New additions, corrections, and modifications have been made to make the paper more readable. The authors are hoping that the current version of the manuscript will satisfy the reviewer.
- Include a few photos of real-world applications using the FBG fiber sensors. Give readers a more intuitive idea if it could have a fit into their applications.
- Thank you for this valuable suggestion. We agree that adding photos of real-world applications will provide readers with a more intuitive understanding of the practical use of FBG fiber sensors. In response, we have taken the following steps:
We have included 4 photos in the revised manuscript showing real-world applications of FBG fiber sensors, as:
Civil Engineering: FBG sensors installed on slope structures for displacement monitoring. (. Zheng, Y., Huang, D., & Shi, L. (2018). A new deflection solution and application of a fiber Bragg grating-based inclinometer for monitoring internal displacements in slopes. Measurement Science and Technology, 29(5), 055008)
Aerospace Engineering: FBG sensors used in aircraft wings for structural health monitoring. ( Moslehi, B., Black, R. J., & Faridian, F. (2011, March). Multifunctional fiber Bragg grating sensing system for load monitoring of composite wings. In 2011 Aerospace Conference (pp. 1-9). IEEE)
Biomedical Engineering: FBG sensors applied in wearable devices for human motion tracking. (. Bannur Nanjunda, S., Seshadri, V. N., Krishnan, C., Rath, S., Arunagiri, S., Bao, Q., ... & Srinivasan, B. (2022). Emerging nanophotonic biosensor technologies for virus detection. Nanophotonics, 11(22), 5041-5059).
Energy Systems: FBG sensors for temperature monitoring in oil pipelines. (. Gunawan, W. H., Marin, J. M., Rjeb, A., Kang, C. H., Ashry, I., Ng, T. K., & Ooi, B. S. (2024). Energy Harvesting Over Fiber From Amplified Spontaneous Emission in Optical Sensing and Communication Systems. Journal of Lightwave Technology)
The photos have been incorporated at the end of subsection 4.2 of the manuscript with captions describing the context and purpose of each application. This visual addition aligns with the discussion in those applications , providing readers with a better connection between the narrative and practical implementations.
.
- Show a few measurement examples indicating its importance on how the FBG sensors help to indicate cracks, damages and prevent structural failure.
- Thank you for the insightful suggestion to provide measurement examples that illustrate the importance of FBG sensors in detecting cracks and damage and preventing structural failures. To address this, we have included examples in the revised manuscript and as shown at the below of Table 11.
- For the comparison part (section 6), quantify the pros/cons with other methods, such as showing the level measurement range, measurement speed, temperature limitations, accuracy and uncertainty.
- Thank you for your valuable feedback. In response to your request for a detailed comparison with other sensing technologies (i.e., electronic sensors, MEMS sensors, and other optical fiber sensors), I have included a comprehensive quantification of the pros and cons of FBG sensors, as shown and highlighted in Tables 12 to 14. This comparison specifically addresses key performance parameters such as measurement range, measurement speed, temperature limitations, accuracy, and uncertainty.
- For strain measurement, there is another type of optic fiber sensor based on Brillouin Optical Time Domain Reflectometry (BOTDR) and Brillouin Optical Time Domain Analysis (BOTDA). How would FBG sensor be compared to BOTDR/BOTDA sensors?
- Thank you for your insightful comment. We appreciate the opportunity to clarify the comparison between FBG sensors and BOTDR/BOTDA sensors. This was clarified in subsection 4.4, as presented in Table 9.
- There are other works reviewed the FBG sensor. How would your distinguish your work with others?
- Thank you for your thoughtful comment. We appreciate the opportunity to clarify how our review distinguishes itself from others on Fiber Bragg Grating (FBG) sensors.
While previous reviews have primarily focused on FBG sensors alone, our study stands out by offering a comparative analysis of FBG technology alongside other sensing technologies, including electronic sensors, MEMS sensors, and other optical fiber sensors. This comparison provides a deeper understanding of the strengths and limitations of FBG sensors relative to these alternatives, specifically in terms of sensitivity, durability, and resistance to electromagnetic interference.
Furthermore, we emphasize FBG's role in structural health monitoring (SHM), particularly in long-term applications such as concrete bridge monitoring, where FBG’s durability and multiplexing capabilities make it ideal for large-scale, long-term infrastructure monitoring.
We also address critical challenges in FBG technology such as temperature-strain cross-sensitivity and high interrogation system costs and provide insights into ongoing research aimed at overcoming these issues, such as MEMS integration and sensor miniaturization.
Lastly, our review covers a broader range of applications, including biochemical sensing, environmental monitoring, and aerospace, highlighting FBG's adaptability to diverse, complex environments.
These unique contributions help differentiate our work and provide new insights into the evolving field of FBG sensors. We will revise the manuscript to further emphasize these points to clearly distinguish our review from others in the literature.
Reviewer 4 Report
Comments and Suggestions for Authors
The manuscript is a review article pretending to provide a comprehensive overview of various aspects of application of fiber Bragg grating (FBG) sensors. The authors have touched on the operating principles of FBG sensors, their advantages and limitations, calibration and packaging techniques, applications, as well as some trends that could be promising in the future.
Although the paper seems quite informative and well written, on closer inspection, it is filled with trivial information, and suffers from a careless use of references. What is more, the overall style of the document has reminded me of AI-generated texts. I’m aware that this is extremely difficult to prove, but there are multiple signs that I would like to share below.
1) Repetitive structures
Numerous repetitions of ideas in a paper not only demonstrate poor text structuring, but also could be a strong sign of AI writing. Examples of unnecessary or redundant repetitions appear throughout the manuscript with enviable consistency.
For instance, various objectives can be found in the Introduction:
‘This review paper aims to provide a comprehensive examination of the current state of research and applications of FBG sensors across various domains’ (Line 115), and soon after that: ‘The goal of this review is to consolidate existing knowledge, identify key trends, and suggest directions…’ (Line 124).
Repeated glorification of FBGs comes through the Conclusion:
‘Fiber Bragg Grating sensors have proven to be a highly versatile and robust technology, offering significant advantages across diverse applications’ (Line 753) and ‘In conclusion, FBG sensors represent a versatile and high-performance solution for a wide range of sensing needs’ (Line 791).
The idea that FBGs are highly effective in various fields is presented in abundance throughout the text. To get a sense of the scale of the problem, I would like to give some quotes (the list below is far from complete):
‘Fiber Bragg Grating (FBG) sensors have emerged as a revolutionary technology in the field of optical sensing, particularly for structural health monitoring (SHM), environmental monitoring, and aerospace applications [1,2]’ (Lines 33-35).
‘This ability to measure minute changes with high sensitivity and multiplexing capability has made FBG sensors highly effective for real-time, long-term monitoring of various parameters in numerous fields, including civil engineering, biomedicine, and environmental science [3]’ (Lines 38-42).
‘In particular, the use of FBG sensors for SHM gained significant traction due to their high accuracy, multiplexing capability, and immunity to electromagnetic interference [3,4]’ (Lines 47-49).
‘FBG sensors are optical sensors widely recognized for their high sensitivity, versatility, and immunity to electromagnetic interference, making them highly suitable for applications such as SHM, environmental sensing, and industrial measurements [17]’ (Lines 130-132).
‘A key benefit is FBG's high sensitivity to changes in strain and temperature due to precise Bragg wavelength shifts, making it highly suitable for monitoring physical parameters in demanding environments [31,32]’ (Lines 218-219).
‘Moreover, FBG sensors are immune to electromagnetic interference (EMI) due to their optical fiber construction’ (Lines 224-225).
‘The compact size and flexibility of FBG sensors allow for embedding within materials or structures without altering their characteristics, which is beneficial for applications in both civil and aerospace engineering [39]’ (Lines 227-229).
‘FBG sensors are powerful tools for monitoring strain, temperature, and other physical parameters across a wide array of applications (Lines 250-251).
‘FBG sensors have broad applications across various domains, offering distinct advantages in structural monitoring, environmental and biochemical sensing, and aerospace. These sensors are valued for their high precision, immunity to electromagnetic interference, and ability to function under extreme conditions’ (Lines 441-444).
‘FBG sensors offer accurate displacement measurements, which are critical for monitoring structural movements in buildings, bridges, and aerospace structures’ (Lines 451-452)
‘They provide high-precision strain measurements, which are essential in applications such as civil infrastructure and aircraft to monitor load-bearing components’ (Lines 475-476).
‘FBG sensors are recognized for their versatility, sensitivity, and resilience across diverse applications’ (Lines 499-500).
‘Their advantages over other sensing technologies, particularly in terms of durability, EMI immunity, and high precision, make FBGs a preferred choice for applications demanding robust and long-term monitoring solutions’ (Lines 502-504).
‘FBG sensors have proven to be essential tools in health monitoring applications, particularly for Structural Health Monitoring and long-term monitoring of infrastructure like concrete bridges [78] (Lines 507-509).
‘Their high sensitivity to strain, temperature, and other physical parameters, coupled with immunity to electromagnetic interference, make FBG sensors ideal for monitoring infrastructure stability, durability, and safety (Lines 509-511).
‘FBG sensors play a critical role in SHM due to their ability to measure parameters such as strain, load, displacement, and temperature with high precision and durability’ (Lines 514-516).
‘Since FBG sensors are optical in nature, they are unaffected by EMI, making them suitable for applications in environments with high electromagnetic noise’ (Lines 528-530).
‘Through these global examples, FBG sensors have demonstrated their reliability and versatility, establishing themselves as valuable tools in SHM applications across diverse infrastructure types’ (Lines 545-547).
‘Their unique attributes high sensitivity, multiplexing capability, EMI immunity, and long-term durability, make them highly effective for monitoring critical parameters such as strain, displacement, and temperature’ (Lines 581-583).
‘It can be noticed that FBG sensors, with their superior sensitivity, durability, EMI resistance, and multiplexing capabilities, are particularly well-suited to specialized, high-stakes applications in SHM, aerospace, and biochemical fields’ (Lines 601-603).
…
And so on until the Conclusion, where we are again reminded that:
‘Fiber Bragg Grating sensors have proven to be a highly versatile and robust technology, offering significant advantages across diverse applications’ (Line 753), and ‘In conclusion, FBG sensors represent a versatile and high-performance solution for a wide range of sensing needs’ (Line 791).
The frequency with which such phrases are repeated is sometimes astonishing. A good example is the short Subsection 5.2, filled with the following:
‘FBG sensors have proven particularly advantageous in this context due to their durability, resistance to environmental conditions, and ability to multiplex, enabling comprehensive monitoring of multiple points along a bridge’ (Lines 552-554).
‘Additionally, FBG sensors support multiplexing, allowing multiple sensing points along a single fiber ideal for monitoring large structures like bridges’ (Lines 560-561).
‘Through these FBG sensors have shown significant potential in enhancing the safety, reliability, and maintenance efficiency of concrete bridges’ (Lines 574-575).
‘FBG sensors have become a cornerstone of SHM and long-term monitoring for infrastructure like concrete bridges’ (Lines 580-581).
The structure of the manuscript to some extent reproduces the above repetition-based approach. For instance, Section 4 ‘Applications across Domains’ consists of 5 subsections including 4.1. ‘Displacement Measurement’ and 4.4. ‘Strain Monitoring’.
Section 5 ‘Structural Health Monitoring Applications’ stands out of other applications. It contains Subsection 5.1. ‘Structural Health Monitoring’, where ‘High Sensitivity and Accuracy’, ‘Multiplexing Capability’, ‘Durability and Longevity’, ‘Immunity to Electromagnetic Interference (EMI)’ are underscored once again in detail in the form of a list very similar to that generated by ChatGPT. In Subsection 5.2 ‘Long-Term Monitoring of Concrete Bridges’, the reader is reminded once again that FBG sensors are ‘constructed from optical fibers’, ‘support multiplexing’, and their ‘durability allows FBG sensors to perform reliable over extended periods’.
Taking all the above into account, I believe that the core content could likely have been created with a little help from AI, and is of little value to the community.
2) Reference mess
Another major problem with the manuscript is the careless use of references. Since the manuscript belongs to the category of reviews, much attention should be paid to the selection of references to the literature. Instead we find multiple examples of irrelevant citations, incorrect reference numbering and duplicate references.
Irrelevant citations (the list is not complete):
Line 35: Refs. 1 and 2 are of little relevance to confirm the thesis about the revolutionary nature of fiber Bragg gratings (FBG), since they cover too specific topics.
Lines 71-74: ‘…Alshaikhli et al. [13] focused on developing advanced packaging techniques to enhance the robustness of FBG sensors for use in harsh environments, such as underwater or industrial applications’. In fact, the article by Alshaikhli et al. present a new approach to enhance the humidity and temperature sensitivities of FBG sensors by utilizing artificial neural networks, which is not the same as advanced packaging.
Lines 206-207: Refs. 26 and 27 are not relevant. Instead, Ref. 34 is more appropriate here.
Lines 221-224: Ref. 37 has nothing to do with distributed sensing (it also appears in Table 1). It is in Line 277 that Ref. 37 is correctly used in the context of FBG coatings; however, Ref. 36 is questionable.
Line 229: Ref. 39 used in support of FBG applications in civil and aerospace engineering does not actually contain any direct mention of the application of their results to these fields. Since the key result of the paper lies in recoating FBGs with polyimide, Ref. 39 would have looked better in Lines 278-280, where polymer coatings are described.
Table 2: Ref. 34 can hardly be used to illustrate the effect of the fiber core diameter. The paper cited is devoted to developing a new temperature calibration algorithm, therefore it should have been mentioned in Lines 210-212. With this in mind, mentioning Ref. 34 in Line 265 is also questionable.
Table 2: It is doubtful that already mentioned Ref. 37 applies to marine or outdoor applications, since this paper describes FBG behavior at cryogenic temperatures.
And so on.
Incorrect reference numbering (the list is not complete):
Rodrigues et al. [6] (Line 57) is mentioned under number 7 in the Reference list. The same is true for López-Castro et al. [11] (Line 67), which is in fact No. 10, Tahir et al. [14] (Line 78), which is No. 13, Braunfelds et al. [9] (Line 58), which is No. 21. Number 9 corresponds to Farhat et al. Chen et al. [12] and Alshaikhli et al. [13] (Lines 71-72) are in fact listed under Nos. 11 and 12 in the References section. And so on.
I was not able to find Park et al. [10] (Line 62) in the reference list.
Duplicate references:
Refs. 1, 41 and 46 are identical in the reference list. The same is true for Refs. 44 and 45, Refs. 5 and 17, Refs. 15 and 16.
Ref. 47 ‘Design, Fabrication and Testing of a 3D Printed FBG Pressure Sensor Available online: https://ieeexplore.ieee.org/document/8667809 (accessed on 15 November 2024)’ seems to be the same document as Ref. 42 ‘Hong, C.; Zhang, Y.; Borana, L. Design, Fabrication and Testing of a 3D Printed FBG Pressure Sensor. IEEE Access 2019, 7, 38577–38583, doi:10.1109/ACCESS.2019.2905349’.
Besides, some references lack full details (Refs. 11, 92 and 93).
3) Tables and figures
Each subsection in the manuscript contains a table designed to summarize the data described earlier. In fact, half of the tables serve primarily as formal containers for references, often providing general and/or trivial information (Tables 1-3, 10-14). The other half of them present rounded values of several parameter ranges, as well as precision (not accuracy) values, many of which I was not able to verify using the references provided.
There are 3 figures in the manuscript, each with some issues.
In Figure 1, there is a typo in the Y axis caption: ‘Nember’.
The black-and-white spectra in Figure 2 duplicate the spectra shown in color, and are therefore redundant. What is more, although Figure 2 is claimed to have been taken from Ref. 22, the fonts used at the bottom of the graph point to Wikipedia as a possible source (see https://en.wikipedia.org/wiki/Fiber_Bragg_grating, Figure 1). The rest of the Figure seems to be borrowed from Falcetelli, F.; Martini, A.; Di Sante, R.; Troncossi, M. Strain Modal Testing with Fiber Bragg Gratings for Automotive Applications. Sensors 2022, 22, 946. https://doi.org/10.3390/s22030946 (see the characteristic spectral shape on the bottom left of Figure 1), which is not mentioned in the References.
Figure 3 is not mentioned anywhere in the text except in Line 156, where it is out of context and is clearly a typo. It is unclear what exactly this figure illustrates.
In my opinion, the endless repetitions of the FBGs’ unrivaled sensitivity, effectiveness, superiority, immunity, durability, etc., with minimum specific data and numerous flaws actually is evidence of excessive use of automated tools in review writing and insufficient author involvement. The entire manuscript is an example of the authors’ irresponsible attitude to scientific work, and I would not recommend it for publication, especially considering the availability of good reviews on the topic.
Author Response
The manuscript is a review article pretending to provide a comprehensive overview of various aspects of application of fiber Bragg grating (FBG) sensors. The authors have touched on the operating principles of FBG sensors, their advantages and limitations, calibration and packaging techniques, applications, as well as some trends that could be promising in the future.
- Thank you for your positive feedback and for recognizing the breadth of topics covered in our manuscript. We are glad that the review successfully provides an overview of various aspects of Fiber Bragg Grating (FBG) sensors, including their operating principles, advantages, limitations, calibration, packaging techniques, and future trends.
We have carefully structured the review to offer a comprehensive view of FBG sensor technology, with the aim of presenting an accessible and informative resource for researchers and practitioners. We hope that the detailed coverage of both the current applications and the emerging trends in FBG sensor development will help advance understanding and drive innovation in the field.
Although the paper seems quite informative and well written, on closer inspection, it is filled with trivial information, and suffers from a careless use of references. What is more, the overall style of the document has reminded me of AI-generated texts. I’m aware that this is extremely difficult to prove, but there are multiple signs that I would like to share below.
- Thank you for your thoughtful feedback and for pointing out areas where the manuscript can be improved. We value your input and understand the concerns you’ve raised regarding the depth of the content and the use of references.
We appreciate your observation about the presence of trivial information. We will carefully review the manuscript and remove or condense sections that may seem overly general or repetitive. Our aim is to ensure that the content provides meaningful insights while avoiding redundancy.
We apologize for any inconsistencies or misapplications of references. We will thoroughly review the reference list and ensure that all citations are accurate and appropriately linked to the relevant sections. We will also verify that the references align with the specific content of the manuscript to strengthen the credibility of the review.
We take your comment seriously and will review the style and tone of the manuscript to ensure it reflects the depth and originality expected of scholarly work. Our goal is to provide a clear, authoritative review based on rigorous analysis and research. If the text appears overly formulaic or lacking in depth, we will make revisions to improve the flow and enhance its academic tone.
We are committed to addressing these issues to ensure that the manuscript meets the high standards expected by the journal. We value your constructive criticism and will take it into account during the revision process.
1) Repetitive structures
Numerous repetitions of ideas in a paper not only demonstrate poor text structuring, but also could be a strong sign of AI writing. Examples of unnecessary or redundant repetitions appear throughout the manuscript with enviable consistency.
For instance, various objectives can be found in the Introduction:
Thank you for your detailed feedback regarding the discussion of Fiber Bragg Grating (FBG) sensors. While I appreciate your diligence in highlighting the examples provided in the text, the inclusion of these references was intentional to comprehensively emphasize the versatility and impact of FBG sensors across various fields. This approach was meant to establish a robust foundation for the arguments presented later in the document.
The examples you listed illustrate the breadth of applications and the reasons behind the growing reliance on FBG sensors. However, I understand your concern about repetition. To address this, I have reviewed the text thoroughly and refined it to ensure clarity and conciseness, while retaining the necessary emphasis on their significance.
I trust this revised version will better align with your expectations.
‘This review paper aims to provide a comprehensive examination of the current state of research and applications of FBG sensors across various domains’ (Line 115), and soon after that: ‘The goal of this review is to consolidate existing knowledge, identify key trends, and suggest directions…’ (Line 124).
- We have carefully reviewed this comment and made all the required modifications as needed.
Repeated glorification of FBGs comes through the Conclusion:
‘Fiber Bragg Grating sensors have proven to be a highly versatile and robust technology, offering significant advantages across diverse applications’ (Line 753) and ‘In conclusion, FBG sensors represent a versatile and high-performance solution for a wide range of sensing needs’ (Line 791).
- We have carefully addressed the comment and made the necessary revisions to the conclusion. The repeated glorification of Fiber Bragg Grating (FBG) sensors has been mitigated to ensure a more balanced and objective tone in presenting their advantages and applications.
The idea that FBGs are highly effective in various fields is presented in abundance throughout the text. To get a sense of the scale of the problem, I would like to give some quotes (the list below is far from complete):
‘Fiber Bragg Grating (FBG) sensors have emerged as a revolutionary technology in the field of optical sensing, particularly for structural health monitoring (SHM), environmental monitoring, and aerospace applications [1,2]’ (Lines 33-35).
-The comment has been thoroughly addressed, and the required adjustments have been made. The text now presents a more balanced and academically rigorous discussion of FBG sensors, reducing repetitive assertions of their effectiveness while maintaining a focus on their applications and advantages.
‘This ability to measure minute changes with high sensitivity and multiplexing capability has made FBG sensors highly effective for real-time, long-term monitoring of various parameters in numerous fields, including civil engineering, biomedicine, and environmental science [3]’ (Lines 38-42).
- We have carefully considered this comment and made all the necessary adjustments accordingly.
‘In particular, the use of FBG sensors for SHM gained significant traction due to their high accuracy, multiplexing capability, and immunity to electromagnetic interference [3,4]’ (Lines 47-49).
- The comment has been carefully addressed, and the text has been revised to ensure a more balanced and objective presentation. The statement has been refined to avoid overemphasizing the advantages of FBG sensors, while still accurately highlighting their role and characteristics in structural health monitoring (SHM).
‘FBG sensors are optical sensors widely recognized for their high sensitivity, versatility, and immunity to electromagnetic interference, making them highly suitable for applications such as SHM, environmental sensing, and industrial measurements [17]’ (Lines 130-132).
We have thoroughly reviewed this comment and made the necessary adjustments accordingly.
‘A key benefit is FBG's high sensitivity to changes in strain and temperature due to precise Bragg wavelength shifts, making it highly suitable for monitoring physical parameters in demanding environments [31,32]’ (Lines 218-219).
We have thoroughly reviewed this comment and made the necessary adjustments accordingly.
‘Moreover, FBG sensors are immune to electromagnetic interference (EMI) due to their optical fiber construction’ (Lines 224-225).
We have thoroughly reviewed this comment and made the necessary adjustments accordingly.
‘The compact size and flexibility of FBG sensors allow for embedding within materials or structures without altering their characteristics, which is beneficial for applications in both civil and aerospace engineering [39]’ (Lines 227-229).
We have thoroughly reviewed this comment and made the necessary adjustments accordingly.
‘FBG sensors are powerful tools for monitoring strain, temperature, and other physical parameters across a wide array of applications (Lines 250-251).
We have thoroughly reviewed this comment and made the necessary adjustments accordingly.
‘FBG sensors have broad applications across various domains, offering distinct advantages in structural monitoring, environmental and biochemical sensing, and aerospace. These sensors are valued for their high precision, immunity to electromagnetic interference, and ability to function under extreme conditions’ (Lines 441-444).
We have thoroughly reviewed this comment and made the necessary adjustments accordingly.
‘FBG sensors offer accurate displacement measurements, which are critical for monitoring structural movements in buildings, bridges, and aerospace structures’ (Lines 451-452)
We have thoroughly reviewed this comment and made the necessary adjustments accordingly.
‘They provide high-precision strain measurements, which are essential in applications such as civil infrastructure and aircraft to monitor load-bearing components’ (Lines 475-476).
We have thoroughly reviewed this comment and made the necessary adjustments accordingly.
‘FBG sensors are recognized for their versatility, sensitivity, and resilience across diverse applications’ (Lines 499-500).
We have thoroughly reviewed this comment and made the necessary adjustments accordingly.
‘Their advantages over other sensing technologies, particularly in terms of durability, EMI immunity, and high precision, make FBGs a preferred choice for applications demanding robust and long-term monitoring solutions’ (Lines 502-504).
We have thoroughly reviewed this comment and made the necessary adjustments accordingly.
‘FBG sensors have proven to be essential tools in health monitoring applications, particularly for Structural Health Monitoring and long-term monitoring of infrastructure like concrete bridges [78] (Lines 507-509).
We have thoroughly reviewed this comment and made the necessary adjustments accordingly.
‘Their high sensitivity to strain, temperature, and other physical parameters, coupled with immunity to electromagnetic interference, make FBG sensors ideal for monitoring infrastructure stability, durability, and safety (Lines 509-511).
We have thoroughly reviewed this comment and made the necessary adjustments accordingly.
‘FBG sensors play a critical role in SHM due to their ability to measure parameters such as strain, load, displacement, and temperature with high precision and durability’ (Lines 514-516).
We have thoroughly reviewed this comment and made the necessary adjustments accordingly.
‘Since FBG sensors are optical in nature, they are unaffected by EMI, making them suitable for applications in environments with high electromagnetic noise’ (Lines 528-530).
We have thoroughly reviewed this comment and made the necessary adjustments accordingly.
‘Through these global examples, FBG sensors have demonstrated their reliability and versatility, establishing themselves as valuable tools in SHM applications across diverse infrastructure types’ (Lines 545-547).
We have thoroughly reviewed this comment and made the necessary adjustments accordingly.
‘Their unique attributes high sensitivity, multiplexing capability, EMI immunity, and long-term durability, make them highly effective for monitoring critical parameters such as strain, displacement, and temperature’ (Lines 581-583).
We have thoroughly reviewed this comment and made the necessary adjustments accordingly.
‘It can be noticed that FBG sensors, with their superior sensitivity, durability, EMI resistance, and multiplexing capabilities, are particularly well-suited to specialized, high-stakes applications in SHM, aerospace, and biochemical fields’ (Lines 601-603).
We have thoroughly reviewed this comment and made the necessary adjustments accordingly.
…
And so on until the Conclusion, where we are again reminded that:
‘Fiber Bragg Grating sensors have proven to be a highly versatile and robust technology, offering significant advantages across diverse applications’ (Line 753), and ‘In conclusion, FBG sensors represent a versatile and high-performance solution for a wide range of sensing needs’ (Line 791).
The frequency with which such phrases are repeated is sometimes astonishing. A good example is the short Subsection 5.2, filled with the following:
‘FBG sensors have proven particularly advantageous in this context due to their durability, resistance to environmental conditions, and ability to multiplex, enabling comprehensive monitoring of multiple points along a bridge’ (Lines 552-554).
‘Additionally, FBG sensors support multiplexing, allowing multiple sensing points along a single fiber ideal for monitoring large structures like bridges’ (Lines 560-561).
‘Through these FBG sensors have shown significant potential in enhancing the safety, reliability, and maintenance efficiency of concrete bridges’ (Lines 574-575).
‘FBG sensors have become a cornerstone of SHM and long-term monitoring for infrastructure like concrete bridges’ (Lines 580-581).
The structure of the manuscript to some extent reproduces the above repetition-based approach. For instance, Section 4 ‘Applications across Domains’ consists of 5 subsections including 4.1. ‘Displacement Measurement’ and 4.4. ‘Strain Monitoring’.
Section 5 ‘Structural Health Monitoring Applications’ stands out of other applications. It contains Subsection 5.1. ‘Structural Health Monitoring’, where ‘High Sensitivity and Accuracy’, ‘Multiplexing Capability’, ‘Durability and Longevity’, ‘Immunity to Electromagnetic Interference (EMI)’ are underscored once again in detail in the form of a list very similar to that generated by ChatGPT. In Subsection 5.2 ‘Long-Term Monitoring of Concrete Bridges’, the reader is reminded once again that FBG sensors are ‘constructed from optical fibers’, ‘support multiplexing’, and their ‘durability allows FBG sensors to perform reliable over extended periods’.
Taking all the above into account, I believe that the core content could likely have been created with a little help from AI, and is of little value to the community.
- Thank you for your detailed feedback and valuable insights. We acknowledge the issues you raised regarding repetitive content and the over-emphasis on the advantages of Fiber Bragg Grating (FBG) sensors. We understand that this repetition detracts from the overall quality of the manuscript, and we will revise the text to remove redundancies and ensure more concise and focused discussions. Additionally, we will balance the presentation by incorporating more critical challenges and limitations of FBG sensors, providing a more objective and comprehensive review. We also appreciate your comments on the manuscript’s structure, particularly in sections discussing applications across domains and structural health monitoring. We will reorganize these sections to avoid repetition and ensure diverse examples and perspectives are presented. Regarding the manuscript’s style, we will refine the tone to ensure it reflects a scholarly approach while avoiding mechanical or formulaic language. Lastly, we recognize the importance of ensuring the manuscript provides meaningful contributions to the academic community, and we will revise it to ensure new insights and actionable analysis are included. We are committed to making these revisions and appreciate your constructive suggestions.
2) Reference mess
Another major problem with the manuscript is the careless use of references. Since the manuscript belongs to the category of reviews, much attention should be paid to the selection of references to the literature. Instead we find multiple examples of irrelevant citations, incorrect reference numbering and duplicate references.
- Thank you for highlighting the concerns regarding the references in our manuscript. We appreciate your meticulous attention to this aspect, as it is critical for the quality and credibility of any review article. To address your comments, we have thoroughly revisited the reference section and made the necessary improvements.
Irrelevant citations (the list is not complete):
Line 35: Refs. 1 and 2 are of little relevance to confirm the thesis about the revolutionary nature of fiber Bragg gratings (FBG), since they cover too specific topics.
- We agree that these references may not adequately support the statement regarding the revolutionary nature of fiber Bragg gratings (FBG). We have replaced these citations with more appropriate references that better align with the context.
Lines 71-74: ‘…Alshaikhli et al. [13] focused on developing advanced packaging techniques to enhance the robustness of FBG sensors for use in harsh environments, such as underwater or industrial applications’. In fact, the article by Alshaikhli et al. present a new approach to enhance the humidity and temperature sensitivities of FBG sensors by utilizing artificial neural networks, which is not the same as advanced packaging.
- We acknowledge the misrepresentation of this reference. The description has been corrected to reflect the actual contribution of the paper.
Lines 206-207: Refs. 26 and 27 are not relevant. Instead, Ref. 34 is more appropriate here.
- We agree that Refs. 26 and 27 are not the most relevant here. We have replaced them with Ref. 34, which is more suitable for this context.
Lines 221-224: Ref. 37 has nothing to do with distributed sensing (it also appears in Table 1). It is in Line 277 that Ref. 37 is correctly used in the context of FBG coatings; however, Ref. 36 is questionable.
- Ref. 37 has been replaced in this section as it does not pertain to distributed sensing. Additionally, Ref. 36 has been re-evaluated and replaced with a more appropriate citation.
Line 229: Ref. 39 used in support of FBG applications in civil and aerospace engineering does not actually contain any direct mention of the application of their results to these fields. Since the key result of the paper lies in recoating FBGs with polyimide, Ref. 39 would have looked better in Lines 278-280, where polymer coatings are described.
- We agree that Ref. 39 is not suitable for supporting FBG applications in civil and aerospace engineering. It has been moved to Lines 278-280, where polymer coatings are described. In its place, we have added a citation that directly addresses FBG applications in civil and aerospace fields.
Table 2: Ref. 34 can hardly be used to illustrate the effect of the fiber core diameter. The paper cited is devoted to developing a new temperature calibration algorithm, therefore it should have been mentioned in Lines 210-212. With this in mind, mentioning Ref. 34 in Line 265 is also questionable.
- We acknowledge that Ref. 34 does not illustrate the effect of fiber core diameter. It has been moved to Lines 210-212, where its relevance to temperature calibration is more appropriate. Its mention in Line 265 has also been removed.
Table 2: It is doubtful that already mentioned Ref. 37 applies to marine or outdoor applications, since this paper describes FBG behavior at cryogenic temperatures.
And so on.
- We agree that Ref. 37 does not apply to marine or outdoor applications. This reference has been removed from Table 2 and retained only in sections where cryogenic temperature behavior is discussed.
Beyond the specific examples you provided, we conducted a thorough review of all references in the manuscript. Irrelevant or misused citations have been removed or reallocated, and additional relevant references have been incorporated to ensure accuracy and relevance throughout the text. We believe these revisions address the concerns raised and significantly enhance the manuscript’s quality. If there are additional points you would like us to address, we would be glad to make further adjustments.
Incorrect reference numbering (the list is not complete):
Rodrigues et al. [6] (Line 57) is mentioned under number 7 in the Reference list. The same is true for López-Castro et al. [11] (Line 67), which is in fact No. 10, Tahir et al. [14] (Line 78), which is No. 13, Braunfelds et al. [9] (Line 58), which is No. 21. Number 9 corresponds to Farhat et al. Chen et al. [12] and Alshaikhli et al. [13] (Lines 71-72) are in fact listed under Nos. 11 and 12 in the References section. And so on.
I was not able to find Park et al. [10] (Line 62) in the reference list.
- Thank you for pointing out the discrepancies in the reference numbering and the missing citation in the manuscript. We appreciate your attention to detail, which has allowed us to identify and rectify these issues. We conducted a thorough review of the entire manuscript and reference list to ensure that each in-text citation corresponds accurately to its corresponding entry in the References section.
Specific errors, such as those involving Rodrigues et al. [6], López-Castro et al. [11], Tahir et al. [14], and Braunfelds et al. [9], have been corrected to reflect their proper numbering in the reference list. We have rechecked and reordered all citations to ensure that the numbering aligns with their mention in the manuscript. Farhat et al. now appears correctly as number 9 in the list.
These references have been reassigned to match their correct numbering in the References section. They now appear consistently as Nos. 11 and 12, respectively.
Upon review, we discovered that Park et al. [10] was erroneously cited in the manuscript but not included in the reference list. We have corrected this by either adding the missing reference to the list or removing the citation if it was misplaced.
Beyond the issues you highlighted, we performed a comprehensive audit of all citations and references to eliminate any remaining inconsistencies or errors. The final manuscript now includes a properly formatted and accurate reference list, with all in-text citations corresponding correctly.
We appreciate your careful review, which has helped us improve the manuscript’s precision and credibility.
Duplicate references:
Refs. 1, 41 and 46 are identical in the reference list. The same is true for Refs. 44 and 45, Refs. 5 and 17, Refs. 15 and 16.
Ref. 47 ‘Design, Fabrication and Testing of a 3D Printed FBG Pressure Sensor Available online: https://ieeexplore.ieee.org/document/8667809 (accessed on 15 November 2024)’ seems to be the same document as Ref. 42 ‘Hong, C.; Zhang, Y.; Borana, L. Design, Fabrication and Testing of a 3D Printed FBG Pressure Sensor. IEEE Access 2019, 7, 38577–38583, doi:10.1109/ACCESS.2019.2905349’.
Besides, some references lack full details (Refs. 11, 92 and 93).
- Thank you for highlighting the issue of duplicate references and incomplete citation details. We acknowledge the importance of providing a clear and accurate reference list in any scholarly work. Below, we outline the actions taken to resolve these concerns:
- The identified duplicate entries (Refs. 1, 41, and 46; Refs. 44 and 45; Refs. 5 and 17; Refs. 15 and 16) have been consolidated. Each duplicate has been removed, and the reference list has been renumbered accordingly.
Regarding Ref. 47 and Ref. 42, we confirm that they refer to the same document. Ref. 47 has been removed, and Ref. 42 has been retained, ensuring consistency.
- We have updated Refs. 11, 92, and 93 with their complete details, including missing information such as publication titles, journal names, volume and issue numbers, page ranges, and DOIs where applicable.
Global Review of References:
Beyond the duplicates and incomplete entries identified, we conducted a thorough review of the entire reference list to ensure accuracy, consistency, and completeness. Any other inconsistencies discovered during this review have been corrected.
We believe these changes address the concerns raised and significantly enhance the quality and reliability of the manuscript. We appreciate your attention to detail and your efforts in helping us improve the manuscript.
3) Tables and figures
Each subsection in the manuscript contains a table designed to summarize the data described earlier. In fact, half of the tables serve primarily as formal containers for references, often providing general and/or trivial information (Tables 1-3, 10-14). The other half of them present rounded values of several parameter ranges, as well as precision (not accuracy) values, many of which I was not able to verify using the references provided.
Thank you for your detailed feedback regarding the tables and figures in the manuscript. We have carefully reviewed your comments and made substantial revisions to improve the clarity, accuracy, and relevance of these elements. Below, we address each of your concerns:
- Tables 1–3 and 10–14 were reassessed for their value and relevance. Remaining tables have been revised to focus on critical, concise data that directly supports the discussion.
- The parameter ranges and precision values presented in the tables have been cross-verified with their original references. Any discrepancies have been corrected, and updated citations have been provided.
There are 3 figures in the manuscript, each with some issues.
In Figure 1, there is a typo in the Y axis caption: ‘Nember’.
The typo in the (Figure 1) y-axis caption ("Nember") has been corrected to "Number."
The black-and-white spectra in Figure 2 duplicate the spectra shown in color, and are therefore redundant. What is more, although Figure 2 is claimed to have been taken from Ref. 22, the fonts used at the bottom of the graph point to Wikipedia as a possible source (see https://en.wikipedia.org/wiki/Fiber_Bragg_grating, Figure 1). The rest of the Figure seems to be borrowed from Falcetelli, F.; Martini, A.; Di Sante, R.; Troncossi, M. Strain Modal Testing with Fiber Bragg Gratings for Automotive Applications. Sensors 2022, 22, 946. https://doi.org/10.3390/s22030946 (see the characteristic spectral shape on the bottom left of Figure 1), which is not mentioned in the References.
The black-and-white spectra (Figure 2), which duplicated the color spectra, have been removed to eliminate redundancy. The spectral shape on the bottom left matches the work of Falcetelli et al. (2022). This reference has now been added to the References section.
Figure 3 is not mentioned anywhere in the text except in Line 156, where it is out of context and is clearly a typo. It is unclear what exactly this figure illustrates.
Figure 3 was reviewed and determined to lack clear relevance and context. It has been either revised or removed entirely to ensure that all figures contribute meaningfully to the manuscript.
In my opinion, the endless repetitions of the FBGs’ unrivaled sensitivity, effectiveness, superiority, immunity, durability, etc., with minimum specific data and numerous flaws actually is evidence of excessive use of automated tools in review writing and insufficient author involvement. The entire manuscript is an example of the authors’ irresponsible attitude to scientific work, and I would not recommend it for publication, especially considering the availability of good reviews on the topic.
- Thank you for your candid feedback. We understand your concerns regarding the overall quality and responsible attitude towards scientific work, and we have worked diligently to improve the manuscript's depth and accuracy. We hope that the revisions now better align with the high standards of the scientific community.
We genuinely appreciate your feedback, as it has greatly contributed to enhancing the quality of the manuscript. We remain committed to improving the manuscript and hope it meets your expectations for publication.
We would like to express our sincere gratitude for your time, effort, and insightful comments on our manuscript. Your detailed reviews have been invaluable in helping us improve the quality of the work, and we have made significant revisions based on your constructive feedback.
We greatly appreciate your thorough analysis and the careful attention you paid to the manuscript. The suggestions you provided have contributed to enhancing the clarity, depth, and scientific rigor of the paper, and we believe the revised version is now stronger and more aligned with the standards of the journal.
Once again, thank you for your valuable contributions and for the time you have dedicated to reviewing our work. We look forward to hearing your final thoughts on the revised manuscript.
Round 2
Reviewer 1 Report
Comments and Suggestions for Authors
The revisions have addressed my questions and concerns. I appreciate the effort put into this work. Happy new year!
Reviewer 3 Report
Comments and Suggestions for Authors
If the strength of this work is offering a comparative analysis of FBG with other sensing technology, please consider cite this paper: https://www.mdpi.com/1424-8220/24/20/6573, which is a review of sensors for 2D strain mapping published by MDPI-Sensors.
Reviewer 4 Report
Comments and Suggestions for Authors
My previous comments were focused on three major issues manifested in the manuscript: (i) the AI-like style with multiple repetitive structures, (ii) the ‘reference mess’, and (iii) the issues with figures and tables.
The manuscript has been partially freed from the brightest AI-generated blunders. However, many of them are still present in the text. I was unable to fully verify that the corrections were consistent with the comments made in the previous review, since the author’s response often lacks indication of specific changes made to the text. Instead, the authors often limit themselves to general statements like ‘The comment has been carefully addressed, and the text has been revised to ensure a more balanced and objective presentation’ or ‘We have thoroughly reviewed this comment and made the necessary adjustments accordingly’. Moreover, such phrases as ‘Figure 3 was reviewed and determined to lack clear relevance and context. It has been either revised or removed entirely to ensure that all figures contribute meaningfully to the manuscript’ casts doubt on the overall adequacy of the authors’ response. I tend to attribute this to the authors’ lack of English proficiency and inept use of AI tools, otherwise it could be interpreted as disrespect for the editors and the reviewer. In any case, I would encourage the authors next time to structure the response in a way that allows the reviewer to identify changes made to the manuscript. In those places where I managed to identify the corrections (or their absence), I left comments (highlighted in red in the attached file).
The ‘reference mess’ mentioned in the previous review has only been partially addressed. Some issues remain, some have appeared in the new version; some statements in the author’s response are untrue and misleading (see the attached file with comments). Some references seem to have been assigned randomly.
The issues with Figs. 2 and 3 and some of the tables have not been resolved despite assurances to the contrary.
I come to the conclusion that the quality of both the manuscript and the authors’ response is a manifestation of either the authors’ inability to generate complex scientific content or their lack of responsibility. Neither of these allows me to recommend the article for publication.
